# Condition-specific genetic interaction maps reveal crosstalk between the cAMP/PKA and the HOG MAPK pathways in the activation of the general stress response

Jenia Gutin[1], Amit Sadeh[1], Ayelet Rahat[1], Amir Aharoni[2] & Nir Friedman[1,*]

## Abstract

Cells must quickly respond and efficiently adapt to environmental changes. The yeast *Saccharomyces cerevisiae* has multiple pathways that respond to specific environmental insults, as well as a generic stress response program. The later is regulated by two transcription factors, Msn2 and Msn4, that integrate information from upstream pathways to produce fast, tunable, and robust response to different environmental changes. To understand this integration, we employed a systematic approach to genetically dissect the contribution of various cellular pathways to Msn2/4 regulation under a range of stress and growth conditions. We established a high-throughput liquid handling and automated flow cytometry system and measured GFP levels in 68 single-knockout and 1,566 double-knockout strains that carry an *HSP12-GFP* allele as a reporter for Msn2/4 activity. Based on the expression of this Msn2/4 reporter in five different conditions, we identified numerous genetic and epistatic interactions between different components in the network upstream to Msn2/4. Our analysis gains new insights into the functional specialization of the RAS paralogs in the repression of stress response and identifies a three-way crosstalk between the Mediator complex, the HOG MAPK pathway, and the cAMP/PKA pathway.

**Keywords** budding yeast; genetic interactions; Msn2 and Msn4; signaling pathways; transcriptional response to stress

**Subject Categories** Chromatin, Epigenetics, Genomics & Functional Genomics; Transcription; Signal Transduction

**Mol Syst Biol. (2015) 11: 829**

## Introduction

Single-cell organisms face an ever-changing environment, where temperature, salinity, pressure, and nutrients, to name a few, can change rapidly. The ability of these organisms to monitor, respond, and adapt to environmental changes is crucial for their survival (Bahn *et al*, 2007; Zaman *et al*, 2008). In the yeast *Saccharomyces cerevisiae*, adaptation involves, among others, dramatic changes in gene expression that affect ~20% of all its genes (Gasch *et al*, 2000; Causton *et al*, 2001; Berry & Gasch, 2008). This massive transcriptional response includes the activation/repression of genes specific to certain stress conditions, as well as genes that are common to all stress insults, known as "the environmental stress response genes" (ESR). The ESR genes are involved in many cellular functions including carbohydrate metabolism, detoxification of reactive oxygen species, cellular redox reactions, cell wall modification, protein folding and degradation, DNA damage repair, fatty acid metabolism, metabolite transport, and vacuolar and mitochondrial functions (Gasch *et al*, 2000). The coordinated activation of hundreds of the induced ESR (iESR) genes is achieved by the two partially redundant transcription factors Msn2 and Msn4. These paralogous zinc finger proteins bind to stress response elements (STRE) at the promoters of iESR genes and activate them (Estruch & Carlson, 1993; Martínez-Pastor *et al*, 1996; Gasch *et al*, 2000; Causton *et al*, 2001; Berry & Gasch, 2008). Many iESR genes and stress-specific genes are co-regulated by Msn2/4 with additional stress-specific transcription factors, such as Hsf1, Sko1, Hot1, Yap1, Gcn4, and Gis1. Multiple studies have investigated the effect of Msn2/4 activation on gene expression under different stresses (Gasch *et al*, 2000; Causton *et al*, 2001; Berry & Gasch, 2008; Capaldi *et al*, 2008) and identified Msn2/4 involvement in many of the fundamental cellular processes such as cell division, cellular aging, and cell metabolism and has made the ESR an attractive model to study functions as cell memory (Berry & Gasch, 2008; Mitchell *et al*, 2009; Guan *et al*, 2012), mRNA–protein relations

1 School of Computer Science & Engineering, Institute of Life Sciences, Hebrew University, Jerusalem, Israel
2 Department of Life Science, National Institute for Biotechnology in the Negev, Ben-Gurion University of the Negev, Be'er Sheva, Israel
 *Corresponding author. Tel: +972 2 549 4557; E-mail: nir@cs.huji.ac.il

(Lee *et al*, 2011), and transcriptional noise (McCullagh *et al*, 2010; Stewart-Ornstein *et al*, 2012; Petrenko & Chereji, 2013).

Fine tuning of the iESR genes expression depends on regulation of Msn2/4 activity at multiple levels (Sadeh *et al*, 2011), including nuclear translocation (Görner *et al*, 1998; Smith *et al*, 1998; Jacquet *et al*, 2003; Gonze, 2008), nuclear hyper phosphorylation (Garreau *et al*, 2000), degradation (Durchschlag *et al*, 2004; Lallet *et al*, 2006), DNA binding (Hirata *et al*, 2003), and alteration of Msn2/4 targets' chromatin structure (Mitchell *et al*, 2008; Sadeh *et al*, 2011). Msn2/4 were found to be regulated by several major signaling pathways including the cAMP/PKA (Boy-Marcotte *et al*, 1998; Görner *et al*, 1998; Garmendia-Torres *et al*, 2007; Lee *et al*, 2008), TOR (Beck & Hall, 1999; Mayordomo *et al*, 2002; Medvedik *et al*, 2007), HOG MAPK (Rep *et al*, 2000; Capaldi *et al*, 2008), SNF1/AMPK (Mayordomo *et al*, 2002; De Wever *et al*, 2005) pathways, as well as by GSK-3 homologs activity (Hirata *et al*, 2003; reviewed by Zaman *et al*, 2008; De Nadal *et al*, 2011; Broach, 2012). Due to the complexity of inputs, it is unclear how these myriad pathways interconnect to process environmental signals and produce a coherent, condition-dependent, timely, and quantitative gene expression output. Therefore, Msn2/4 are prime candidates for studying complex cellular output in response to complex input (Fig 1A).

A classical strategy to understand functional output of a complex system is genetic interaction analysis that can uncover the functional consequences of each gene in the network, pathway structure, and interactions between pathways (Schuldiner *et al*, 2005). Large-scale genetic interactions studies that measure changes in total fitness (e.g., colony size/growth rate) illustrate the power of such strategy (Schuldiner *et al*, 2005; Costanzo *et al*, 2010; Ryan *et al*, 2012). However, fitness is a fairly blunt phenotype, which integrates multiple effects, and does not focus on specific mechanisms. Moreover, many perturbations have subtle defects that are undetectable using such coarse phenotypes (Breslow *et al*, 2008). An alternative approach is to use gene expression profiles as an information-rich phenotype (Capaldi *et al*, 2008). This phenotype can distinguish different cellular effects of the perturbations (Kemmeren *et al*, 2014), but more difficult to acquire and are therefore typically restricted to a relatively small number of perturbations.

Here, we have combined the high-throughput power of an automated mating (synthetic genetic array, SGA) methodology (Tong, 2004) with a readout of a quantitative expression phenotype to study the pathways and mechanisms that regulate the iESR activity under different types of stress. We utilized a prototypical Msn2/4-regulated reporter gene (*HSP12-GFP* allele) (Martínez-Pastor *et al*,

1996; Causton *et al*, 2001; Lallet *et al*, 2004; Erkina *et al*, 2008; Sadeh *et al*, 2011) and measured its activity in different genetic backgrounds and different stress and growth conditions. We combined these phenotypes with measurement of the nuclear localization of Msn2 to elucidate the Msn2/4-regulating network. Our results confirm prior description of a complex regulatory network, where the iESR is regulated by many genes with condition-specific contributions. Using data from double and triple mutants, we reconstructed quantitative genetic interaction and epistasis maps of the iESR-regulating network under three environmental stress conditions and two growth conditions. This analysis highlights the significance of transcriptional repression relief in the activation of the iESR and identifies functional specialization of the RAS paralogs. Based on the inter-pathway interactions in our data, we suggest the existence of a crosstalk between the Mediator complex, the HOG MAPK pathway, and the cAMP/PKA pathway. This crosstalk is mediated by the Ras1 protein and maintains the repression of stress-responsive genes in optimal conditions.

## Results

### Establishing fluorescence reporter assay for assaying general stress response

To quantitatively measure the general stress response in real time, we adopted a reporter gene approach (Fig 1B). Following previous works (Martínez-Pastor *et al*, 1996; Causton *et al*, 2001; Lallet *et al*, 2004; Karreman & Lindsey, 2005; Erkina *et al*, 2008; Sadeh *et al*, 2011), we used a C-terminally GFP-tagged Hsp12 fusion protein as our reporter. Expression of Hsp12 (Sales *et al*, 2000) is a sensitive and robust sensor for multiple types of stress, and an integral part of the iESR (Gasch *et al*, 2000; Causton *et al*, 2001; Berry & Gasch, 2008; Sadeh *et al*, 2011). In line with previous experiments at both the mRNA (Gasch *et al*, 2000; Causton *et al*, 2001; Neuert *et al*, 2013) and the protein level (Hasan *et al*, 2002; Sadeh *et al*, 2011), we verified Hsp12-GFP to be a highly sensitive stress reporter, with a wide dynamic range. *HSP12* is also induced during late log phase through the diauxic shift into early stationary phase (Fig EV1A). The expression of *HSP12* is largely Msn2/4 dependent (Sadeh *et al*, 2011), with little response in the Δ*msn2*Δ*msn4* double-knockout strain under a variety of conditions (Fig EV1B). The residual Hsp12-GFP induction in Δ*msn2*Δ*msn4* strain suggests that additional factors can activate *HSP12*, although their contribution is smaller

---

**Figure 1. Dissecting pathways regulating the general stress response in five conditions.**

A The transcription factors Msn2/4 are the main regulators of the induced branch of yeast general stress response. Msn2/4 are regulated by multiple pathways that respond to a large range of environmental conditions. The motivation for this work was to systematically dissect the contribution of the pathways to the general stress response and reconstruct the regulatory network upstream to Msn2/4.

B Experimental design. Strains containing genetic perturbations and genomically integrated *HSP12-GFP* reporter were stimulated. The GFP levels of each strain were measured in 2–3 repeats using flow cytometry under three stress and two growth conditions (Materials and Methods).

C Examples of target gene deletions that affect the levels of Hsp12-GFP under various stress conditions. The median levels of Hsp12-GFP relative to WT are presented. The knockouts can increase/decrease the GFP levels significantly and this effect can be general or stress specific.

D The median levels of Hsp12-GFP in 68 mutant strains under five conditions, shown relative to the matching WT levels (log ratio, note color scale; Δ marks are omitted for succinctness). The rows and the columns are clustered hierarchically. The clustering highlights the strong activators/repressors of *HSP12-GFP* and highlights groups with condition-specific effects.

E Knockouts were classified according to their condition specificity (Materials and Methods). (left) Venn diagram showing the specificity/commonality of knockouts to stress conditions. (right) Venn diagram comparing stress-dependent effects (in one or more stress conditions) to growth effects.

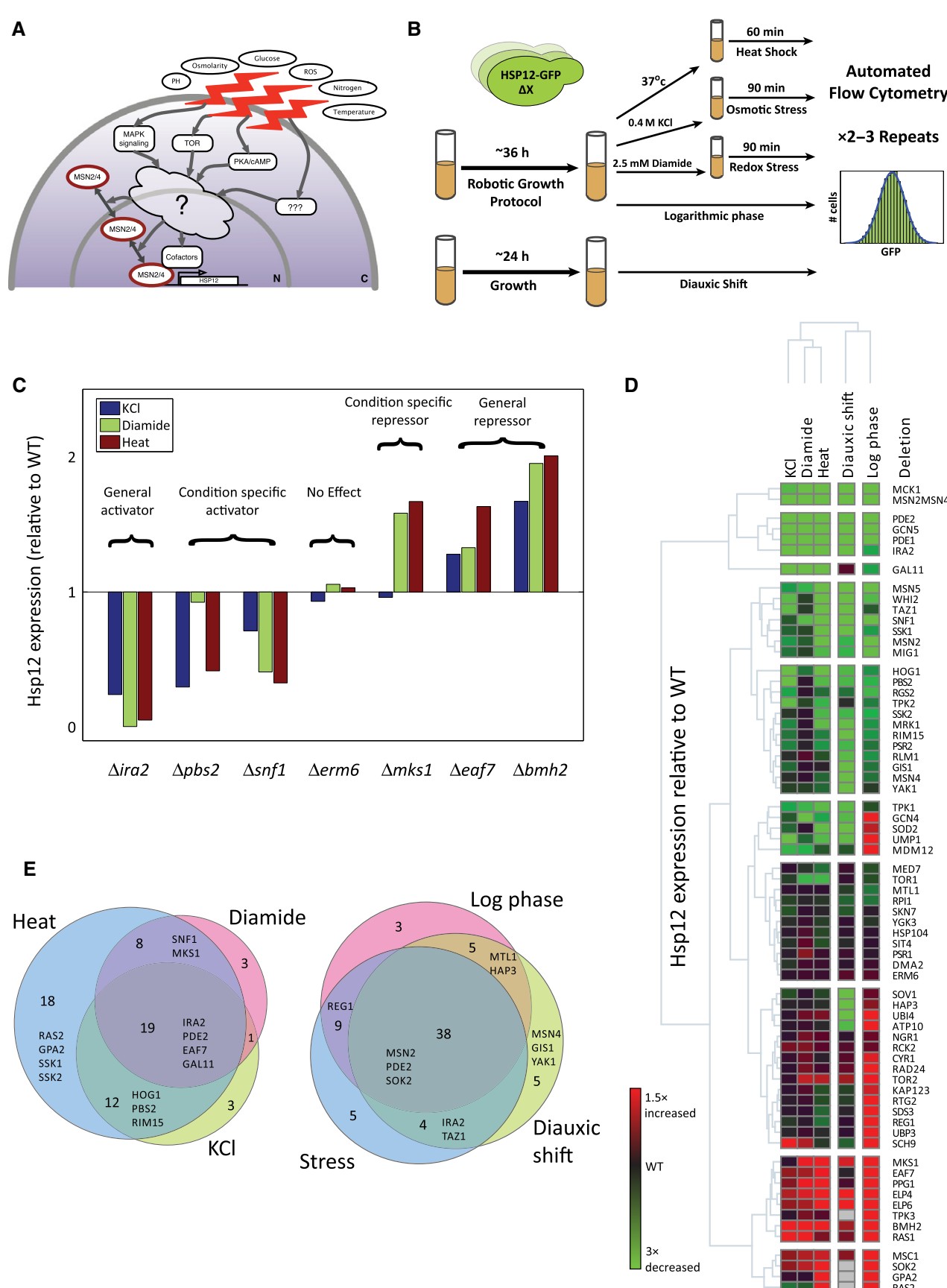

**Figure 1.**

than 20% of the wild-type induction response. One exception to this rule is the response to KCl stress, where ~30% of the induction is Msn2/4 independent, consistent with the known activity of the HOG pathway in *HSP12* induction (Capaldi *et al*, 2008).

Next, we devised an automated protocol for assaying stress response in multiple strains (Materials and Methods). This 96-well format protocol uses automated adaptive dilution to ensure that cells from multiple strains with varying growth rates are grown to mid-log phase with basal Hsp12-GFP levels and sufficient number of cells in each well. Once grown, the cells were then stimulated with a variety of stresses (Materials and Methods), and automatically transferred to a flow cytometer to measure GFP fluorescence values (Fig 1B, Materials and Methods). Comparing Hsp12-GFP fluorescence levels in biological replicates of multiple deletion strains performed over a period of several months shows excellent agreement at log phase ($R = 0.98$, Fig EV2A) and after induction ($R = 0.98$, Fig EV2B), confirming the robustness and reproducibility of this system.

### The general stress response is induced by a combination of generic and stress-specific pathways

To dissect the regulatory networks regulating the general stress response, we focused on a target set of genes. We selected genes based on either known genetic/physical interactions with Msn2/4 (according to online interaction databases; Stark *et al*, 2006; Szklarczyk *et al*, 2011) or their relations to different stress response and signaling pathways (Table EV1). Of the 97 knockout strains of selected genes, several were extremely sick and others showed repeated suppression of constitutive activation of the stress pathways (Lang *et al*, 2013). These were removed from the analysis, leaving 68 target genes. We constructed strains carrying the stress reporter *HSP12-GFP* allele with single knockouts of the selected target genes (Materials and Methods). For three essential genes, we used DAmP alleles (Yan *et al*, 2008). See Table EV2 for a detailed genotype list.

We examined these strains under a variety of conditions. Three different stress conditions were chosen: heat (shift from 30°C to 37°C for 60 min), redox stress (2.5 mM diamide for 90 min), and osmotic stress (0.4 M KCl for 90 min). Time in each stress condition was selected based on time-course run for WT and representative knockout strains. In addition, two growth conditions were assayed: mid-log growth and post-diauxic shift. We find that the deletions of many genes have significant effects on Hsp12-GFP levels under these conditions, identifying both positive and negative regulators of *HSP12* expression (Fig 1C and D, Table EV3). These genes encode proteins belonging to various cellular pathways/processes, including signal transduction pathways (cAMP/PKA, TOR, HOG, AMPK), protein degradation, protein disaggregation, mitochondrial function, GSK-3 homologs, and chromatin remodeling/modifying proteins.

While *HSP12* is induced under all stress conditions, its mechanism of activation can vary between conditions. We assigned condition-specific effects by comparing the magnitude of the effect for a knockout under different conditions (Materials and Methods). For example, Δ*mks1* has an effect specific to diamide and heat stress (Fig 1C). Summarizing these over all the tested genes, we find that while many genes have a common effect on *HSP12-GFP* expression under all conditions, a subset of genes show condition-specific

response (Fig 1E). For example, deletions of adenylate cyclase activators (Δ*ras2*, Δ*gpa2*) have a much stronger effect under heat stress compared to KCl and diamide. Alternatively, deletions of the MAPK and the MAPKK of the HOG pathway (Δ*hog1*, Δ*pbs2*) have weaker effect in diamide than in KCl and heat stress.

Conversely, when we compare stress-dependent effects, in one or more stress conditions, to growth conditions (mid-log and post-diauxic shift), we find some condition-specific effectors, but a significantly larger common core. However, some knockouts are specific to growth conditions. For example, the deletion of the post-diauxic shift-specific transcription factor Gis1 (Pedruzzi *et al*, 2000) has a more substantial effect in post-diauxic growth.

Notably, some deletions lead to opposite effects under different conditions. For example, deletion of the Mediator complex subunit Gal11 abolishes *HSP12-GFP* expression under all conditions (Fig EV3A), except for post-diauxic shift where it induces higher levels of Hsp12-GFP. In contrast, deletion of several nuclear-encoded mitochondrial genes (*MDM12*, *ATP10*, *HAP3*, *SOV1*) reduces Hsp12-GFP levels in post-diauxic shift when the mitochondria are actively respiring and it induces Hsp12-GFP levels in mid-log growth (Fig EV3B). Other example involves knockout strains exhibiting slow growth. These deletions are of genes from different pathways: *SOD2*—response to reactive oxygen species, *UMP1*—proteolysis, *GCN4*—regulation of amino acid synthesis. In these strains, mid-log levels of Hsp12-GFP are higher than WT, yet, they have lower Hsp12-GFP response to stress. Apparently, those knockouts induce non-optimal growth conditions, which are manifested by slower growth rate and activation of stress response during exponential growth and reduced ability to respond to rapid environmental changes.

### Msn2/4 activity is a central integration point for the general stress response from multiple pathways

Next we asked what is the role of Msn2/4 in the effects we observe in Fig 1D. For example, the effect might require the activity of Msn2/4 (Msn2/4-dependent effect), it might bypass Msn2/4 (Msn2/4-independent effect), or it might be due to cofactor interactions with Msn2/4 (cooperative effect). Finally, the gene might be involved in multiple pathways acting on *HSP12-GFP* and thus represent a mixture of the above scenarios (Fig 2A). To quantify the contribution of Msn2/4 dependence of each gene, we examined GFP expression in triple mutants in which Msn2/4 are deleted in addition to one more modulator (Δ*geneX*Δ*msn2*Δ*msn4*). The effect of each knockout under specific stress condition can be put on a spectrum between fully Msn2/4-independent activity, through Msn2/4-dependent activity, to cooperativity with Msn2/4 (Fig 2B, Materials and Methods). For most tested genes, deletion of Msn2/4 attenuated their effect on *HSP12-GFP* expression, suggesting an Msn2/4-mediated activation. There are, however, notable exceptions. For example, upon KCl stress, deletion of Hog1 aggravates the effect of *MSN2/4* deletion (Fig 2A). This is in agreement with the idea that Hog1 activates *HSP12* by two pathways, one through Msn2/4 and one by directly affecting other factors such as Sko1 and Hot1 (Proft & Struhl, 2002; Capaldi *et al*, 2008). Another example is Ras1, which under the same stress condition strongly represses *HSP12-GFP*, partially independent of Msn2/4.

                    

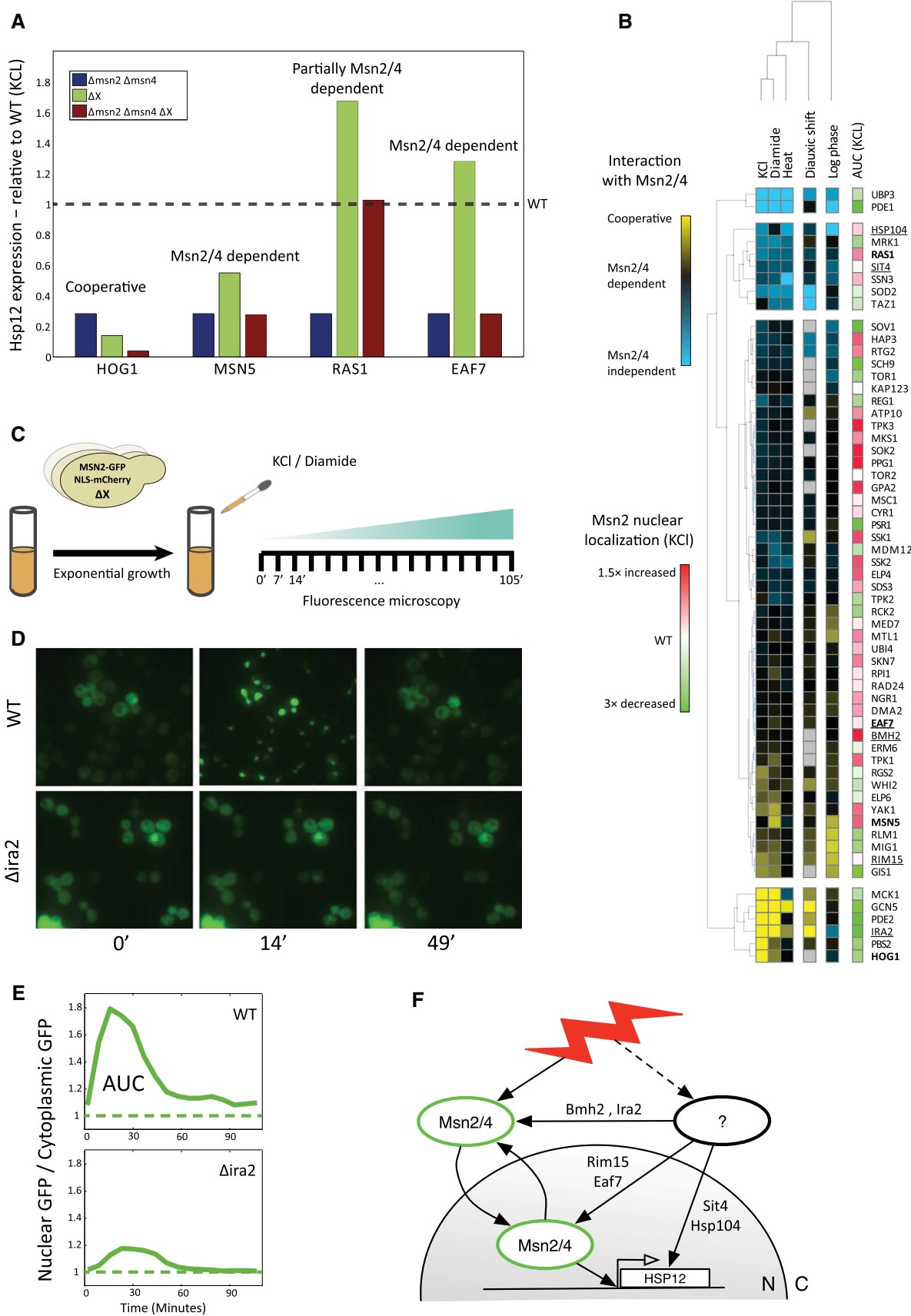

**Figure 2.**

◀

**Figure 2.  Evaluating Msn2/4-dependent activity of each response.**

A   Comparing the Hsp12-GFP levels in Δ*X* vs. Δ*X*Δ*msn2/4* identifies different groups of knockouts. When the effect of Δ*X*Δ*msn2/4* is similar to Δ*msn2/4*, we define Δ*X* as Msn2/4 dependent. When Δ*X*Δ*msn2/4* > Δ*msn2/4*, Δ*X* has an Msn2/4-independent component. Finally, when Δ*X*Δ*msn2/4* < Δ*msn2/4*, *X* and Msn2/4 work cooperatively to activate *HSP12-GFP*.
B   The Msn2/4 dependence of single-gene perturbations. The effects of each knockout on Hsp12-GFP levels were put on a scale between Msn2/4-independent through Msn2/4-dependent activity to cooperativity with Msn2/4 (Materials and Methods). The perturbations and conditions are clustered hierarchically. Most of the knockouts have Msn2/4-dependent effects on Hsp12-GFP levels, and some of those effects can be explained by Msn2 localization. The effect of each knockout on the localization of Msn2-GFP in KCl stress is shown in the rightmost column.
C   Experimental setup of Msn2-GFP localization experiment. We generated strains with a knockout of a single gene and a *ADH1p-MSN2-GFP* plasmid. Using fluorescence microscopy, we followed the localization of Msn2-GFP in those strains in intervals of 7 min after the exposure to KCl and diamide stresses.
D   Msn2-GFP localization after the exposure to 0.4 M KCl. Msn2-GFP is mostly cytoplasmic at the beginning of the experiment (time = 0). In the WT strain, we observe strong nuclear accumulation after ~15 min, while in Δ*ira2* strain this nuclear accumulation is abolished. Nuclear GFP returns to its initial levels ~50 min after the stress.
E   We calculate the median ratio between the nuclear and the cytoplasmic GFP over all the cells in each image (Materials and Methods). Localization changes of Msn2-GFP are clearly visible by monitoring this ratio over time. We estimate the area under the curve in those graphs (AUC) and use this quantity as a measure for the amount of time Msn2-GFP spent in the nucleus in each mutant.
F   Interim summary. The effect of each knockout on the activation of *HSP12-GFP* can be Msn2/4 dependent or independent. The Msn2/4-dependent effects can be mediated by Msn2/4 localization or by other mechanisms that affect Msn2/4 in the nucleus. So far we identified the contribution of each of our knockouts to those different paths of *HSP12-GFP* activation.

Although most of the knockouts show Msn2/4-dependent activity under all tested conditions, this analysis allows us to identify and separate mutations that activate *HSP12-GFP* induction via Msn2/4-independent pathways that are repressed in the WT strain. These might include other activators such as Hot1 (Capaldi *et al*, 2008), Gis1 (Pedruzzi *et al*, 2000), and Hsf1 (Imazu & Sakurai, 2005) as well as removal of repression by repressors such as Sko1 and Sok2 (see below). We thus conclude that Msn2 and Msn4 are the main, but not sole, activation branch of the general stress response.

## Nuclear localization is necessary but not sufficient for Msn2/4 activity

Nuclear localization of Msn2/4 is well established as a major regulation point for their transcriptional activity. During mid-log growth, Msn2/4 are localized to the cytoplasm. Upon stress induction, Msn2/4 are rapidly imported into the nucleus, bind to target genes promoters, and induce their expression (Mayordomo *et al,* 2002; Durchschlag *et al*, 2004; Petrenko & Chereji, 2013; Elfving *et al*, 2014). In the nucleus, Msn2/4 can undergo nuclear export and/or degradation (Durchschlag *et al*, 2004; Santhanam *et al*, 2004; Lallet *et al*, 2006). As a result, stress conditions lead to one or more bursts of Msn2/4 nuclear localization. This pulsatile behavior provides a quantized mechanism for tuning the stress response to the severity and duration of the stress (Hao *et al*, 2013; Petrenko & Chereji, 2013).

To assay the contribution of Msn2/4 localization to their transcriptional output, in various mutant backgrounds, we monitored by time-lapse microscopy the nuclear localization of a constitutively expressed and GFP-tagged Msn2 under osmotic (KCl) and redox (diamide) stresses (Fig 2C and D, Table EV3, Materials and Methods). Prior to stress induction, Msn2-GFP is predominantly cytoplasmic in most genetic backgrounds with the exception of deletion of the nuclear exporter Msn5 (Görner *et al*, 2002; Durchschlag *et al*, 2004) (Fig EV4A). WT cells treated with KCl exhibit rapid (< 10 min) nuclear localization of Msn2-GFP peaking at ~15 min, followed by nuclear export and return to basal levels at ~45 min (Fig 2D). In contrast, deletion of *IRA2* abolishes nuclear import under the same insult (Fig 2D). To quantify the extent of nuclear localization in

each mutant background, we examined time-lapse sequence of cells from the mutant. For each cell, we compute the ratio of nuclear over cytoplasmic GFP intensity over time. We average this quantity over all cells in the field to define the area under the curve (AUC) of a strain (Hao & O'Shea, 2012) (Fig 2E).

Combining the Msn2/4 dependence analysis and Msn2 localization datasets (Fig 2B) provides important clues regarding the mode and location of action of a protein. For example, deletion of *EAF7*, a subunit of the NuA4 histone acetyltransferase complex, increases the levels of Hsp12-GFP in Msn2/4-dependent manner (Figs 1D and 2B) without affecting Msn2/4 nuclear localization pattern (Fig 2B). The data recapitulate the repressive effect of the NuA4 complex on *HSP12* (Mitchell *et al*, 2008) and is consistent with the nuclear localization of Eaf7 (Breker *et al*, 2014). This suggests Eaf7 is involved shutting off transcription following Msn2/4-dependent induction, or alternatively, Eaf7 deletion has an indirect effect on translational efficiency of *HSP12-GFP* transcripts. In contrast, while deletion of *RIM15* also has almost no effect on nuclear localization of Msn2, it reduces Hsp12-GFP levels under osmotic stress (Figs 1D and 2B). Rim15 can phosphorylate Msn2 (Lee *et al*, 2013), and evidence suggests that Rim15 is localized to the nucleus under stress conditions (Pedruzzi *et al*, 2003). Combined with our observations, it is possible that activation of *HSP12-GFP* by Rim15 is through enhancing Msn2 activity in the nucleus. Another example is the deletion of *MSC1*, a gene of unknown function, which also does not affect nuclear localization of Msn2. Combined with the observation of a sharp Msn2/4-dependent increase in Hsp12-GFP levels in Δ*msc1* strain suggests that Msc1 serves to repress (possibly indirectly) Msn2 following its nuclear import.

The response to diamide had different dynamics compared to KCl treatment (Fig EV4B); Msn2-GFP was translocated to the nucleus within ~10 min of diamide addition, at a much lower amplitude than upon KCl stress. Additionally, nuclear Msn2-GFP did not return to basal levels for at least 100 min following exposure to diamide stress. The constant nuclear localization of Msn2 reflects the inability of the cells to adapt to this stress condition. Examining mutant strains, we observe correspondence between the effect of the knockout on nuclear localization (AUC) following diamide and KCl treatments (Fig EV4C). Notable exceptions to this trend are KCl-specific effects of the HOG pathway (Δ*hog1*, Δ*pbs2*),

                    

corresponding to the stress specificity observed in *HSP12-GFP* induction (Fig 1E).

Measurements of Msn2 nuclear localization after both KCl and diamide stresses show that localization is dependent on multiple pathways. In addition to the expected effects of cAMP-regulated import (Görner *et al*, 1998; Garmendia-Torres *et al*, 2007) and nuclear export (Msn5 (Görner *et al*, 2002; Durchschlag *et al*, 2004)), we also observe effects of the SAGA transcriptional initiation complex (Gcn5), transcription factors (Sok2), GSK-3 kinase (Mck1), and degradation pathways (Ubp3). Some of these effects might involve secondary feedback loops, although the constitutive expression of Msn2-GFP in our system is insensitive to transcriptional feedback on Msn2. Thus, these results suggest that Msn2-GFP localization depends on the overall balance between multiple nuclear and cytosolic processes.

Summarizing our results so far, we can place *HSP12* regulators on a scale that combines their effect in an Msn2/4-independent manner, through Msn2/4 localization, and alternative Msn2/4 activation (Fig 2F). Some regulators are involved in more than one type of effect. For example, Hog1 has effects through both Msn2/4 localization and Msn2/4-independent pathways.

**Identifying key decision nodes using epistasis analysis**

Our results up to this point support the notion that the general stress response depends on multiple pathways. To better understand how these effects are mediated and where are the contact points between different sensing pathways, we generated double mutants to uncover genetic interactions. For example, if most of Msn2/4 activity depends on cAMP levels, we would expect that different cAMP/PKA pathway genes will be epistatic over other genes—that is, the effect of the double knockout on Hsp12-GFP expression levels will phenocopy the effect of a deletion of the cAMP/PKA pathway component.

To systematically map genetic interactions, we generated a "mini epistatic map" (Schuldiner *et al*, 2005; Collins *et al*, 2007) of 30 query strains against 56 target strains (Fig 3A, Materials and Methods). The resulting strains contained two mutant alleles and the *HSP12-GFP* reporter gene. In total, the library contained 1,566 strains. We next analyzed Hsp12-GFP levels, as described above (Fig 1B), in these strains under three stress conditions and in mid-log and post-diauxic shift growth conditions (Fig 3B, Materials and Methods, Table EV3). In general, the double-deletion response is consistent with the single-deletion measurements (Fig 1). Comparing the average effect of a mutant across all its corresponding double mutants is in agreement with the effect of the single mutant (Fig EV5A).

Evaluating Hsp12-GFP levels in double-deletion strains allows the identification of genetic interactions. The most intuitive and extensively used type of genetic interaction is *epistasis* (also referred to as *complete epistasis* and *masking epistasis*). Briefly, if the phenotype of a mutation in gene *X* is masked by the addition of a mutation in *Y*, we would say that *Y* is epistatic over *X* and predict that *X* is situated upstream to *Y* in the pathway. We note that this is the original use of the term epistasis and is more restrictive than its use as denoting any genetic interaction, see below. The analysis of epistasis interactions is commonly used to infer pathway structure. For example, examining the interactions of Ira2, Ras2, and Pde2 in response to heat shock, we see that Δ*ras2* is epistatic over Δ*ira2*,

while Δ*pde2* is epistatic over Δ*ras2*, matching the known pathway structure (Fig 3C).

To systematically evaluate epistatic relations, we defined criteria for epistasis (Materials and Methods) and mapped all epistatic relations under all five conditions tested (Table EV3). Out of 825 pairs showing epistatic interactions under at least two stress conditions, 707 are consistent across stress conditions (Fig 3D). An inconsistent example is the epistasis of Δ*ras2* over Δ*ira2* in heat stress and the opposite epistasis in observed under KCl and diamide stresses. This reversal is consistent with the change in Δ*ras2* phenotype under these conditions (Fig 1D).

Those epistasis maps clearly identify key perturbations that are epistatic over many of the others. These include Δ*msn2*Δ*msn4*, corresponding to the observation that most of the effects of the single knockouts in our dataset are Msn2/4 dependent (Fig 2B), and deletions that activate the cAMP pathway (Δ*ira2*, Δ*pde2*, Δ*whi2*) supporting the view of cAMP pathway as key decision integration for the general stress response. However, they also underscore the importance of other components: The HOG pathway, which is usually associated with osmotic stress (KCl), is also a key activator in other measured conditions; Sch9, a TOR/SNF-regulated kinase that can regulate Rim15 (Pedruzzi *et al*, 2003; Wanke *et al*, 2005) and Sko1 (Pascual-Ahuir & Proft, 2007); Sok2, a transcriptional repressor; Gal11, a Mediator component; and Mck1, a GSK-3 kinase with less understood role, which was suggested to regulate Msn2/4 activity (Hirata *et al*, 2003).

**Genetic interactions capture Msn2/4-dependent and Msn2/4-independent effects**

The definition of epistasis, which relies on semi-arbitrary thresholds, is limited to pairs of perturbations where at least one has a large effect, and is most notable when the two single perturbations have opposite effects (e.g., Δ*ras2* vs. Δ*ira2*, Fig 3C). Thus, we aimed to define a quantitative and more general measure of genetic interactions (Schuldiner *et al*, 2005; Collins *et al*, 2007; Costanzo *et al*, 2010). Classically, genetic interactions are defined by comparing the measured phenotype (Hsp12-GFP levels) in the double knockout with the expected phenotype when there is "no interaction" between the two perturbations (Fig 4A, no interaction). For example, both Δ*msn4* and Δ*mig1* reduce Hsp12-GFP levels in response to diamide stress (relative to WT response). The combined deletion Δ*mig1*Δ*msn4* has a phenotype that corresponds to the combination of both defects, and thus, we would say that there is no interaction between the two perturbations. In contrast, while both Δ*msn2* and Δ*msn4* confer phenotype defects, their combination, Δ*msn2*Δ*msn4*, has much lower levels than those calculated by simple combination, hence constituting a synthetic sick phenotype, consistent with their partial redundancy (Fig 4A, negative interaction). In the opposite direction, we see that both Δ*snf1* and Δ*rim15* decrease Hsp12-GFP levels, yet Δ*snf1*Δ*rim15* is close to WT Hsp12-GFP levels, indicating a strong positive interaction (Fig 4A, positive interaction).

The interpretation of interactions crucially depends on the definition of the expected outcome in the neutral case. Different models for the expected outcome were previously proposed, including additive, multiplicative, and other models (Elena & Lenski, 1997; Segrè *et al*, 2005; Mani *et al*, 2008). Fitting each of these models to the data, we explain 42–55% (depending on the condition) of the

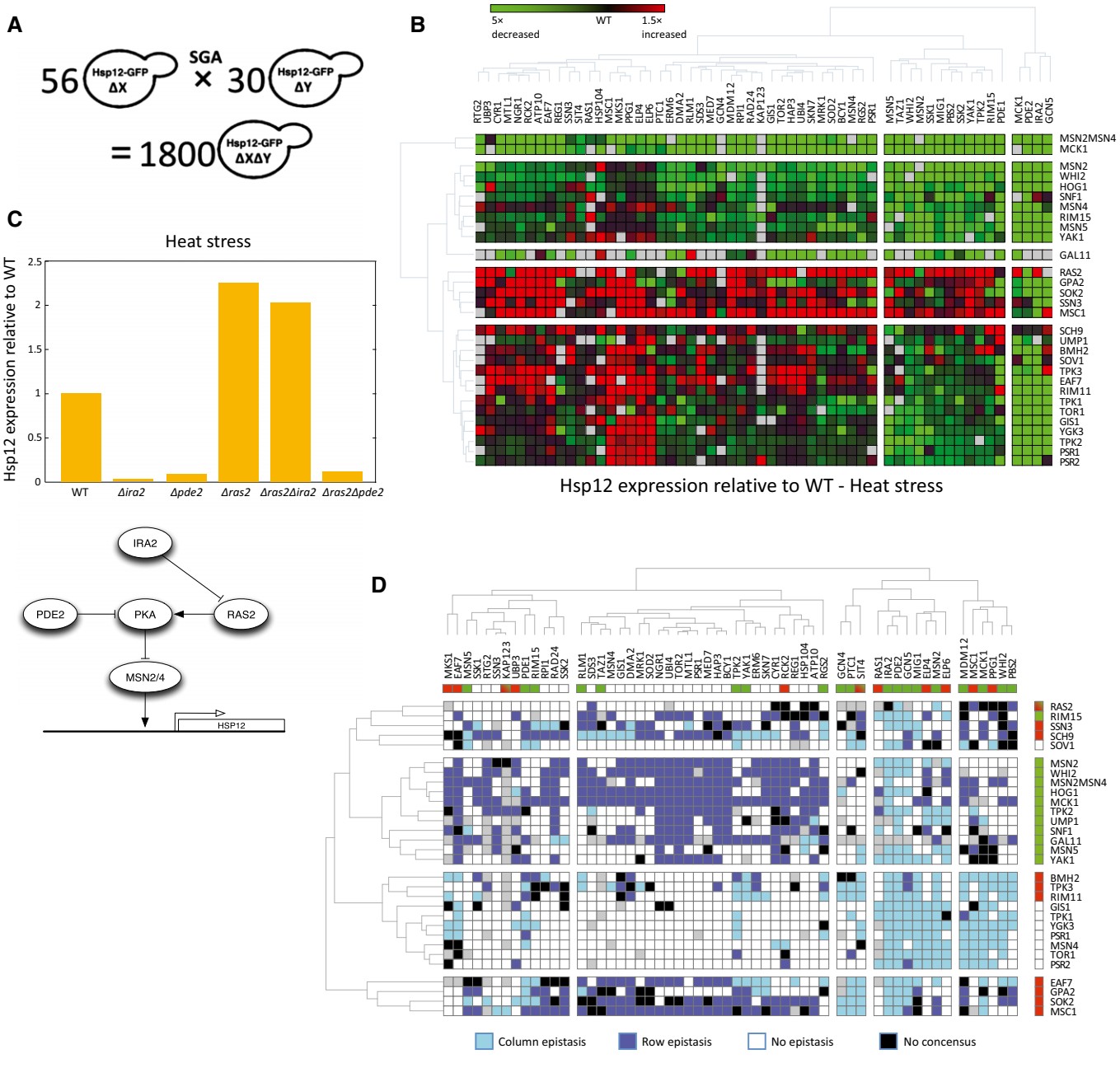

Figure 3. Using double-knockout library to uncover pathway structure.

A   Using the SGA methodology, we constructed a double-knockout library of 30 * 56 strains. Each one of the strains contains deletions of two genes and the *HSP12-GFP* reporter.

B   The library was screened under five conditions (Fig 1B). The raw data of *HSP12-GFP* expression (log) after the exposure to heat stress is shown. Overall, the average effect of the double perturbation resembles the measured effect of the single perturbation.

C   Epistatic interactions in the data that match the well-established structure of the cAMP/PKA pathway. *HSP12-GFP* expression after heat stress is shown. Δ*ras2* is epistatic over Δ*ira2*, implicating that Ira2 is upstream of Ras2. Δ*pde2* is epistatic over Δ*ras2*, implicating that Ras2 is upstream of Pde2.

D   Consensus matrix of the epistasis interactions in our library. We identified all the epistatic interactions in three stress conditions (Materials and Methods). We color double perturbations in which the epistasis observed in at least two of the stress conditions. Row epistasis defined as the epistasis of the gene denoting the row over the gene denoting the column. Column epistasis is vice versa. Double perturbation pairs in which the direction of the epistasis interaction has changed between the conditions are colored in black. The upper row and the rightmost column present the single perturbations effects (red/green is increased/decreased HSP12-GFP levels relative to WT).

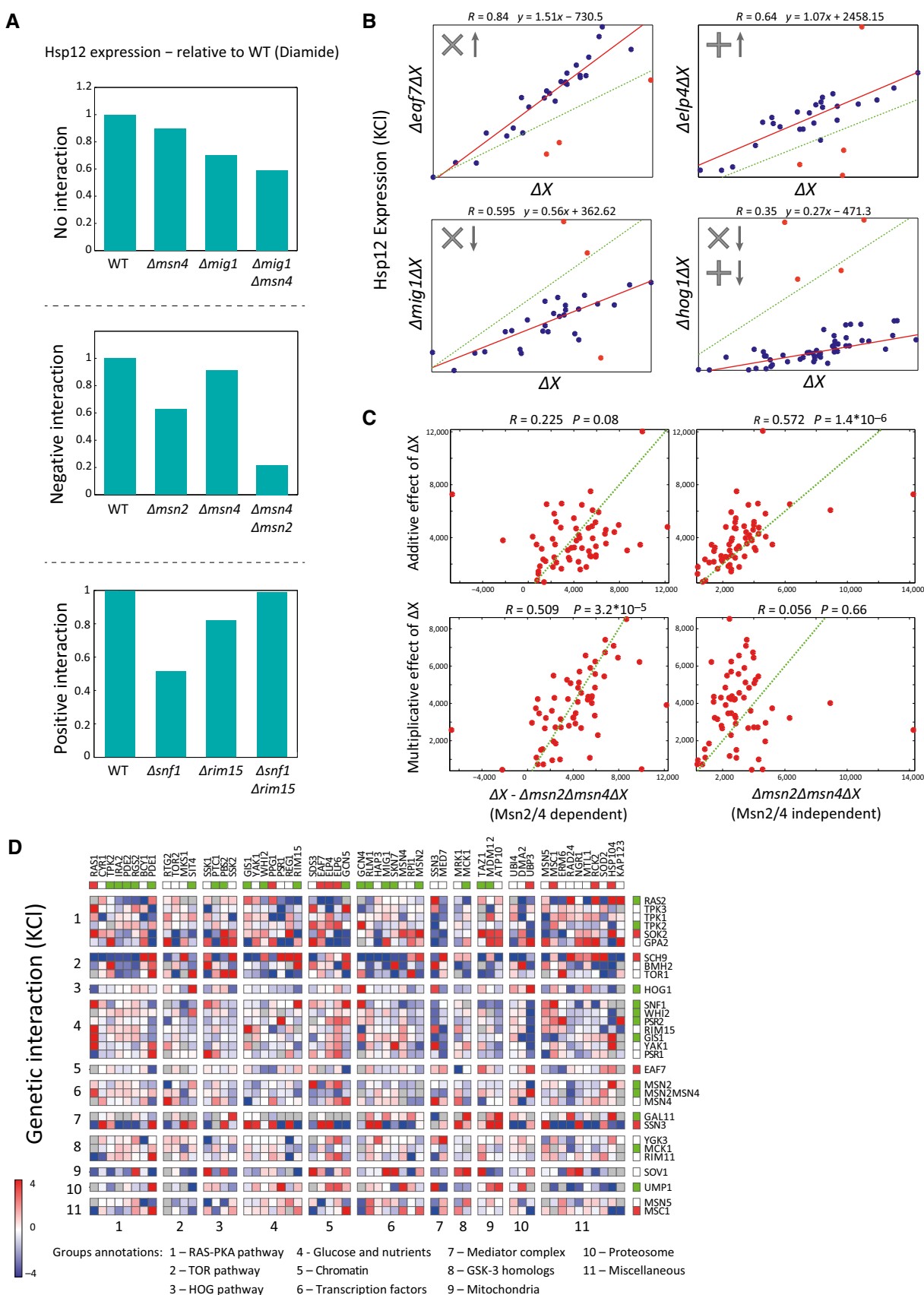

**Figure 4.**

**Figure 4. Calculating genetic interactions.**

A   Examples of different types of genetic interactions from our data. The data show *HSP12-GFP* expression after the exposure to diamide. There is no interaction between two genes if the observed value of the double knockout is equal to the value we would expect to see based on the single-knockout values. If the observed value is higher than the expected, the interaction is positive. If it is lower, the interaction is negative.

B   We observe multiplicative, additive, and combined effects of knockouts on the levels of Hsp12-GFP. Each scatter shows the expression values of *HSP12-GFP* (KCl) in the single knockouts (*x*-axis) vs. the matching double knockouts (*y*-axis). The deviation of the red line from the diagonal (dashed green line) represents the general effect of the single knockout. Points located remotely from the red line are genetic interactions (colored in red).

C   Our combined interaction model associates each single perturbation to additive and multiplicative components. Here, we show that the Msn2/4-dependent effects (KCl) of the perturbations (left) correlate to their multiplicative components (bottom) and their Msn2/4-independent effects (KCl) correlate to their additive components. The opposite does not hold.

D   The genetic interaction map in KCl stress. The interaction value shown is proportional to the difference between the observed and the expected values (based on the dual interaction model). Positive interactions are shown in red and negative in blue. The upper row and the rightmost column present the single-knockout effects (red/green = increase/decrease *HSP12-GFP* levels relatively to WT). The knockouts are sorted by pathways, highlighting inter- and intra-pathway interactions.

variance (Fig EV5B, Materials and Methods). However, closer examination shows that some perturbations interact additively, while others behave multiplicatively. For example, deletion of Elp4 led to a consistent additive increase in Hsp12-GFP levels in most double-knockout strains (Fig 4B). On the other hand, deletion of Eaf7 (component of the NuA4 histone acetyltransferase complex) shows clear multiplicative increase by 50% in most double-knockout strains. We thus considered a combined model where each perturbation has both additive and multiplicative effects (Materials and Methods). This two-parameter model has a significantly better fit to the data explaining overall additional ~10% of the variance (Fig EV5B).

We reasoned that the interaction profile of a perturbation, multiplicative or additive, emanates from the mode of action of the perturbed gene relative to the other genes. For example, above we hypothesized that Eaf7 is involved in shutting down transcription following Msn2/4-dependent transcriptional initiation. This hypothesis is consistent with the multiplicative effect of Eaf7's deletion—presumably, each initiation event generates more transcripts leading to multiplicative amplification in various genetic backgrounds and conditions. To examine this phenomenon more broadly, we compared the estimated multiplicative and additive parameters to other properties of the respective perturbation. We find that the multiplicative parameters correlate ($R = 0.51$, $P = 3.2 * 10^{-5}$—KCl) with the Msn2/4-dependent effect of the perturbation. Conversely, the additive parameters correlate ($R = 0.57$, $P = 1.4 * 10^{-6}$—KCl) with the Msn2/4-independent effect of the perturbation (Fig 4C). Importantly, the opposite does not hold, the multiplicative (additive) parameters are uncorrelated with Msn2/4-independent (dependent) effects (Fig 4C). These striking relations hold in heat, KCl, and post-diauxic shift conditions (Fig EV5C). Note that the model does not take into account Msn2/4 dependency. Thus, the correlation suggests that the mode of interaction estimated by the model captures an inherent property of the network.

Applying our definition of interactions to the data results in intricate interaction maps (Fig 4D, Table EV3). While genetic interaction scores and epistasis are related, they are not identical. For example, we observe strong interactions that are not epistatic (Fig 4A, Δsnf1 vs. Δrim15). Conversely, our definition of expected interaction leads to cases where the expectation is epistatic relations. For example, when one perturbation has a near-zero multiplicative effect and the other has negligible additive effect, we will expect the first perturbation to be epistatic over the second. Indeed, examination of genetic interaction scores between epistatic pairs

uncovers many examples of clear epistasis with close to neutral interaction score (Fig EV6). Due to this partial overlap, we use both in our further analysis.

## Interactions within the cAMP/PKA signaling pathway recapitulate pathway structure and indicate functional specialization of RAS paralogs

The cAMP/PKA is a key cellular homeostasis pathway, playing a key role in regulating stress responses and growth. High cAMP levels, found in growing cells, lead to PKA-mediated phosphorylation of Msn2/4, which inhibits its nuclear localization (Görner *et al*, 1998, 2002).

Focusing on the interactions and epistasis data under heat stress, where this pathway has strong effects (Fig 5A), reconstructs all known aspects of the pathway pertaining to our query genes. Deletion of enhancers of cAMP production such as Ras2 leads to Msn2/4 activation, while deletion of inhibitors of cAMP production such as Ira2 has the opposite effect. However, Δira2 effect requires Ras2, with epistasis of Δras2 over Δira2 (Figs 3C and 5A). This is consistent with the function of Ira2 in repressing Ras2 activity. Similarly, we observe that Δpde2 (high-affinity phosphodiesterase, which reduces AMP levels by converting it to AMP) leads to repression of Msn2/4 and low Hsp12-GFP levels. Moreover, Pde2 is downstream to Ras2 and Δpde2 is epistatic over Δras2, suggesting that even without Ras2-mediated cAMP production, cAMP is produced by alternative pathways. One such alternative is Gpa2-mediated cAMP production, which is responsible for the activation of the adenylate cyclase in response to glucose (Colombo *et al*, 1998). Indeed, Δgpa2 has the strongest effect on Hsp12-GFP levels in mid-log growth and less noticeable effect during stress. In most conditions, we do not observe epistasis of Δgpa2 over the deletion of its repressor Rgs2 (in contrast to Δras2 over Δira2), suggesting additional roles for Rgs2 in this pathway.

Combining prior literature with our observations (see also Table EV4) results in the pathway structure shown in Fig 5B. This reconstruction highlights unappreciated aspects of the pathway. The Sok2 protein is a known target of PKA and has been known to be a transcriptional repressor (Ward *et al*, 1995; Shenhar & Kassir, 2001). Our results show that Δsok2 induces higher levels of Hsp12-GFP even when *MSN2/4* are deleted (Figs 1D and 2B). These observations suggest that Sok2 actively represses *HSP12* transcription, regardless of the activation pathway. Moreover, Δira2 and Δpde2 are both epistatic over Δsok2, supporting PKA-dependent activation of Sok2 (Ward *et al*, 1995; Shenhar & Kassir,

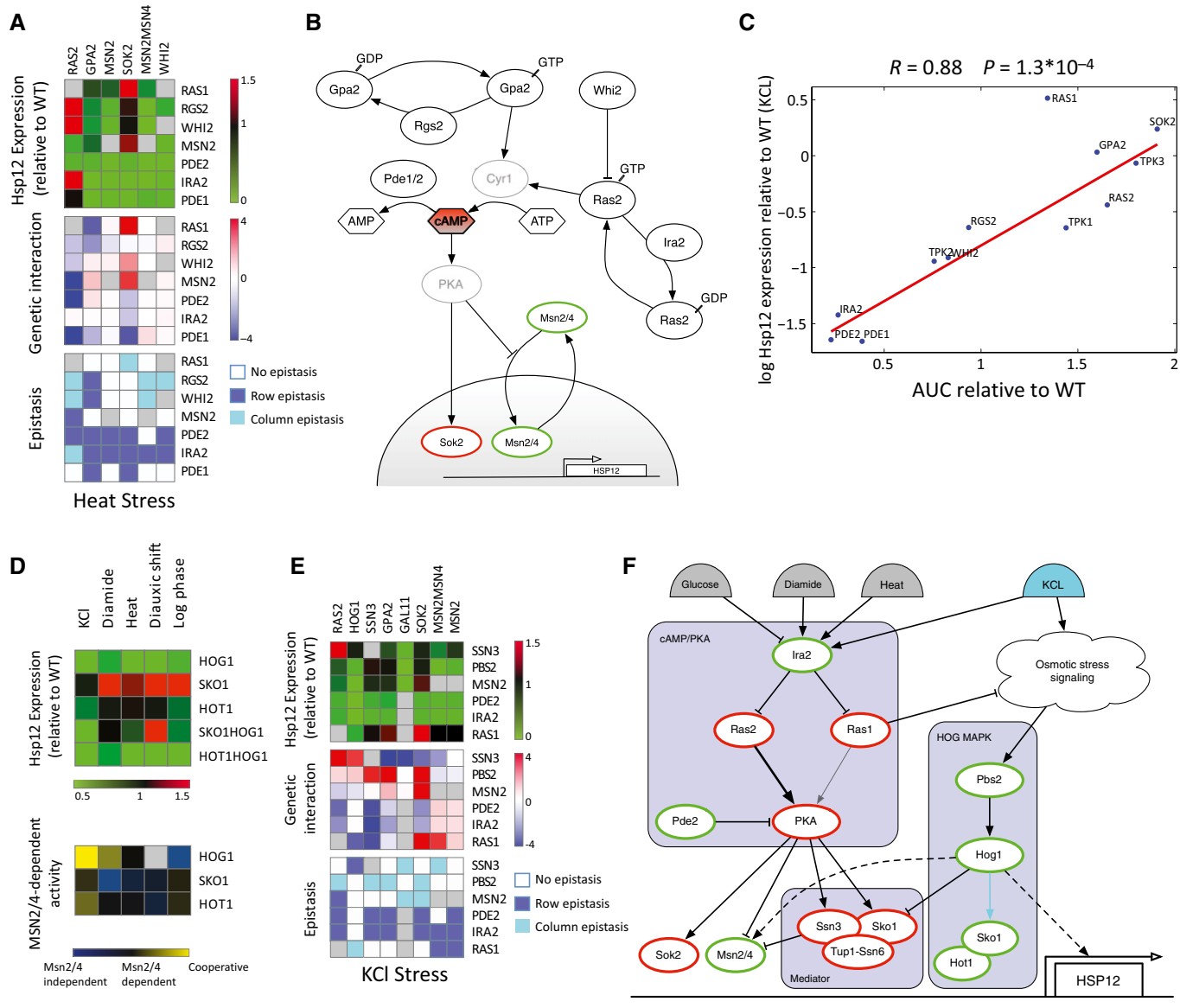

**Figure 5. Analysis of inter- and intra-pathway interactions.**

A    Intra-pathway raw data, genetic interactions, and epistasis interactions of the cAMP/PKA pathway in heat stress. See caption of Fig 3D for row and column epistasis.

B    Reconstruction of the cAMP/PKA pathway based on our data and the literature. Node border color designates the effect of the gene on *HSP12* expression (red and green for repressor and activator, resp.) according to the analysis of Fig 5A.

C    The levels of Hsp12-GFP after KCl stress are highly correlated with nuclear accumulation of Msn2-GFP in the same condition in knockouts of genes from the cAMP/PKA pathway. The only notable exception to this trend is the deletion of Ras1, supporting its Msn2/4-independent effect (Fig 2B) and suggesting that this effect is cAMP independent.

D    HSP12-GFP expression relative to WT in single and double mutants of the HOG MAPK pathway in five conditions (top). Msn2/4-dependent effects of HOG MAPK mutants in five conditions (bottom).

E    Inter-pathway raw data, genetic interactions, and epistasis interactions of the Mediator complex and the cAMP/PKA, HOG MAPK pathways in KCl stress. See caption of Fig 3D for row and column epistasis.

F    Reconstruction of three-way crosstalk between the cAMP/PKA pathway, the HOG MAPK pathway, and the Mediator, based on our data and the literature. Node border color as in Fig 5B.

2001). Thus, the PKA pathway affects *HSP12-GFP* in parallel through both repression of Msn2/4 nuclear import and activation of Sok2-repressive function.

When examining the interactions of Δ*ras2*, we see that it is epistatic over both Δ*ira2* and Δ*whi2* (in heat shock and growth conditions) and that Δ*ira2* is epistatic over Δ*whi2*. This picture

might suggest a linear pathway (Whi2-> Ira2-> Ras2). However, another interpretation, which is also consistent with prior literature, is that both Ira2 and Whi2 repress Ras2, the first by converting Ras2-GTP to Ras2-GDP and the other by shuttling Ras2 to vacuolar degradation (Leadsham *et al*, 2009). In both deletions (Δ*ira2* and Δ*whi2*), there is an increase in Ras2 activity, but we hypothesize

that the deletion of *IRA2* is sufficient to reach saturating levels of Ras2 activity, and thus, Δira2 is epistatic over Δwhi2.

*Saccharomyces cerevisiae* has two paralogous RAS proteins, Ras1 and Ras2, that diverged in the whole-genome duplication event (Byrne & Wolfe, 2005; Wapinski *et al*, 2007). Previous literature suggests that Ras2 is approximately 10-fold more abundant in mid-log growth (Ghaemmaghami *et al*, 2003) and dominant in regulating cAMP levels (Broek *et al*, 1985; Toda *et al*, 1985), suggesting distinct roles for the two paralogs. However, the synthetic lethality of Ras1 and Ras2 suggests some level of redundancy between the two or activity in parallel redundant pathways that support an essential function. Using our data, we were able to clearly distinguish between these two paralogs in terms of Hsp12-GFP induction. Deletion of *RAS2* results in strong *HSP12-GFP* induction in heat shock, mid-log and post-diauxic shift, but it has a mild negative effect in diamide and KCl stresses. In contrast, Δras1 induces *HSP12-GFP* in all conditions, but much less prominently in heat shock and post-diauxic shift (Fig 1D). We also observed that Δmsn2 is epistatic over Δras2 in all conditions, which is consistent with Ras2's role in cAMP production leading to decreased Msn2/4 activity. In contrast, Δras1 has significant Msn2/4-independent effect in all conditions (Fig 2A), and strong positive interactions with Δmsn2 in diamide and KCl stresses, resulting in WT or higher Hsp12-GFP levels. Taken together, our results suggest functional specialization between the two RAS paralogs during stress response that affect the response through distinct partially overlapping pathways.

To further understand the specialization of Ras1/2, we examined whether their effects are mediated through cAMP levels. A good proxy for cAMP activity is the nuclear localization of Msn2. When we compare nuclear import against Hsp12-GFP levels of knockouts of different cAMP/PKA pathway genes, we see a striking correlation (Fig 5C), in contrast to the behavior of other perturbations (Fig EV7). This correlation confirms that the effect of the deletions of the pathway's components on Hsp12-GFP is indeed mediated through the localization of Msn2. Notably, the only exception to this trend is Δras1, with modest increase in Msn2 nuclear localization and substantial induction of Hsp12-GFP. This suggests that Ras1 has a repressive effect on Hsp12-GFP through an alternative pathway which is Msn2/4 independent and apparently also independent of cAMP levels.

### Inter-pathway interactions identify RAS crosstalk with the HOG MAPK pathway, and the Mediator complex

To better understand the functional differences between Ras1 and Ras2 and the pathway they modulate, we examined our interaction/epistasis maps. We observe several strong positive interactions between components of the HOG signaling pathway (Δhog1 and Δpbs2) to the cAMP/PKA pathway (Δgpa2, Δsok2, and Δras2) and to Δssn3, a cyclin-dependent protein kinase that is part of the Mediator complex (Figs 3D and 4D).

Previous literature shows that in response to osmotic stress, the Hog1 MAPK targets the transcription factors Msn2/4, Sko1, and Hot1 leading to the activation of stress-responsive genes including *HSP12* (Rep *et al*, 2000; Proft & Struhl, 2002; Alepuz *et al*, 2003; Capaldi *et al*, 2008; Cook & O'Shea, 2012). We observe an effect of Δhog1 on Hsp12-GFP levels under all conditions, which is only partially Msn2/4 dependent (Fig 2A). Thus, we performed genetic

dissection of Hog1 with Sko1 and Hot1 and Msn2/4 (Fig 5D). We observe that Hot1 is mainly important in KCl stress with an epistasis of Δhog1 over Δhot1. Both are consistent with Hog1-dependent Hot1 activation (Rep *et al*, 2000; Alepuz *et al*, 2003). Moreover, Hot1 activity in KCl stress is not mediated by Msn2/4, but rather cooperative with Msn2/4 consistent with previous reports (Capaldi *et al*, 2008).

Consistent with observations that Hog1 converts Sko1 from a repressor to activator (Proft & Struhl, 2002), we see that Δsko1 increases Hsp12-GFP levels in all conditions except KCl stress. In KCl, we see a mild effect, suggesting that compensation by the strong activation of Hot1 and Msn2/4 cancels Δsko1 effects. Examining Hsp12-GFP levels in a Δhog1Δsko1 strain shows that in KCl, Δhog1 is epistatic over Δsko1 and that in other conditions, Hog1 effect is only partially mediated by Sko1 de-repression. In these other conditions, the effect of Δsko1 is partially Msn2/4 independent (Fig 5D). The emerging picture is that in KCl stress, strong Hog1 activity does not require Sko1, but in other conditions, Hog1 has an effect both by de-repressing Sko1 and by alternative mechanisms. These alternative mechanisms are mediated by Msn2/4 or through direct interactions with the transcriptional machinery (De Nadal *et al*, 2004; Proft *et al*, 2006; Cook & O'Shea, 2012).

Sko1 represses its targets by recruiting the Tup1/Cyc8-repressive complex (Garcia-Gimeno & Struhl, 2000; Proft *et al*, 2001). Tup1/Cyc8-mediated repression of Hsp12 is in cooperation with Ssn3 (Green & Johnson, 2004) (also known as Cdk8/Srb10), a cyclin-dependent kinase that functions in the CDK8-repressive module of the Mediator (Borggrefe *et al*, 2002), suggesting that Δsko1 effects might be mediated by Ssn3. Indeed, examining the interactions between Ssn3 and HOG pathway, we observe strong positive interactions and in some cases epistatic interactions between Δssn3 and either Δhog1 or Δpbs2 in all conditions (Fig 5E). These results suggest that Sko1-independent effects of Δhog1 are also dependent on Ssn3 activity. Given the repressive function of Ssn3, it is possible that these interactions are through indirect effects of Δssn3 rather than its role in transcriptional induction by Hog1-activated transcription factors. One possible mechanism is Ssn3-dependent degradation of Msn2 (Bose *et al*, 2005).

Next, we examined the genetic interactions between HOG, Ssn3, and the cAMP/PKA pathway in KCl stress, where the HOG pathway has a pronounced role (Fig 5E). The deletions of strong activators of the cAMP/PKA pathway (Δras2, Δgpa2) and cAMP-dependent repressor Δsok2 are epistatic over Δpbs2. This observation suggests that the inactivation of the cAMP/PKA pathway can abolish or compensate for the Sko1-dependent repression. The epistasis of Δras2 and Δgpa2, but not Δsok2, can be explained by the direct targeting of Sko1 by PKA (Proft *et al*, 2001).

Surprisingly, in contrast to Δras2, which is epistatic over the HOG pathway (Δpbs2), we observe that Δhog1 is epistatic over Δras1 under all conditions. This suggests that the Msn2/4-independent and cAMP-independent effects of Ras1 are through repression of the HOG pathway. Consistent with that, we observe strong negative interactions between Ras1 and Ssn3 (the end effector of the HOG in most conditions). Moreover, we observe positive interactions across the two arms of the cAMP/PKA and HOG pathway—Δras2 with Δssn3, and Δras1 with Δsok2 suggesting compensatory roles of these two arms in the repression of *HSP12-GFP*.

Additional interaction between the cAMP/PKA pathway and the Mediator complex is through PKA activation of Ssn2, a structural component of the repressive CDK8 module (Chang *et al*, 2004). Thus, the repressive CDK8 module is activated by the cAMP/PKA pathway through both the cAMP branch and alternative branches (e.g., repression of HOG pathway). The strong epistasis of Δ*ira2* over all other mutations in stress conditions (except Δ*ras2* in heat stress) is consistent with Ira2's function in repressing both RAS paralogs. Deleting *IRA2* leads to strong activation of both Ras1 and Ras2. Consequently, there are high cAMP levels resulting in cytosolic sequestration of Msn2/4 and activation of Sok2 and CDK8 module both strongly repressive of *HSP12-GFP*. In addition, Ras1 activity represses the HOG pathway, and blocks alternative activation of *HSP12-GFP* by Sko1 de-repression. We summarize our findings based on these observations and additional ones in a comprehensive model of Msn2/4 activation (Table EV5) in Fig 5F.

## Discussion

This study presents a detailed evaluation of the transcriptional regulation of the general stress response pathway in yeast. We have evaluated the effect of 68 single-gene deletions and 1,566 double deletions in five controlled conditions in replicates. Analysis of this rich dataset uncovered new insights on additional components, interactions, and modes of action of these pathways and corroborated many isolated previous reports in the literature.

### Experimental conditions in studying stress response pathways

*HSP12* is an extremely sensitive sensor for stress. It is one of the first genes to respond and it increases from essentially 0 transcripts/cell to close to 40/cell within few minutes (Neuert *et al*, 2013). As such, Hsp12-GFP levels respond to small environmental changes during growth protocols. To identify effects of gene deletions, we must be certain that differences we observe are not due to experimental procedures. Thus, we have put much effort in establishing an automated growth protocol that leads to reproducible experimental results (Fig EV2A and B). Key elements of this protocol include the following: strict temperature control; long exponential growth prior to experiment without further dilution or OD measurements during the growth period; uniform treatment to all strains, regardless of their growth rate; and finally reaching a narrow range of mid-log population density at the onset of the experiment (Materials and Methods).

### Hsp12 fluorescence activity reporter provides a direct phenotype with high precision and a large dynamic range

Many of the pathways we dissected here have received close scrutiny in various contexts, such as growth, aging, adapting to nutrient depletion, and environmental stress. A key decision in our experimental design was to focus on *HSP12* transcriptional activation. This allowed us to partially disentangle some of the indirect global effects of mutations (such as cell size and growth rate) from ones directly involved in responding to the environmental cues. In contrast to growth phenotypes (Schuldiner *et al*, 2005; Roguev *et al*, 2008; Costanzo *et al*, 2010), the activity reporter provides precise measurements on a large dynamic range. The precision showed

reproducible effects of knockouts in different genetic backgrounds distinguishing between relatively small differences in protein levels (Fig EV2A–C). Consequently, we can detect genetic interactions and epistasis on a wide range of mutations from strong repressors to strong activators.

A potential complication in the analysis of our results might be due to post-transcriptional events that affect Hsp12-GFP levels, such as changes in mRNA stability and translation rates. Our genetic dissection focused on signaling and transcriptional proteins. This reduces the likelihood of post-translational effects, although clearly there might be cross-effects, for example, between stress signaling and translation (Ingolia *et al*, 2009). With that said, most of results are consistent with effects on transcriptional regulation and are also consistent with other observations, such as Msn2 nuclear localization.

### Genetic interactions with an activity reporter phenotype

At the outset of the project, we aimed to use genetic interactions to uncover the structure of the response pathway. To our surprise, simple notions of "neutral" interaction, such as additive or multiplicative effects, were in clear mismatch with our results (Figs 4B and EV5B). We introduced the combined model that captures both additive and multiplicative effects of each mutant and showed that it was better matched to the observations. Moreover, the two types of effects were correlated with independently measured quantities (Fig 4C) increasing our confidence in these measures.

The use of such interaction model raises several questions. We can think of multiplicative effects as representing the effect of mutants along a pathway. Each step in the pathway modulates the output of the previous step and thus has a multiplicative effect. Such a situation is potentially relevant in other experimental investigation of response pathways. In contrast, additive effects are ones that lead to spurious transcription of *HSP12* not through the main signal transduction. The introduction of a richer interaction model also separated quantitative interaction score (the deviation of the double mutant from the expected value) from the qualitative notion of epistasis. For example, a strong repressive mutation is expected to override the effect of other mutations, and thus, it will be epistatic over these other mutations even though the interaction score is small. Consequently, we gain insight not only from the genetic interactions and epistasis maps, but also from the parameters we fit in the neutral model. Taking this idea one step further, one can consider hierarchical interaction models where mutations within sub-branches have multiplicative effects, and between branches have additive effects. Such a model might have fewer deviations and thus capture the structure of the pathway.

### The role of Msn2/4 in *HSP12* activation

We set out to examine Msn2/4 activity levels in stress response. The choice of *HSP12* was based on the large body of literature that uses it as a case study for stress-responsive transcription. Although *HSP12-GFP* response is by and large Msn2/4 dependent (Fig EV1B), there are additional transcription factors that modulate this response. Our analysis also covered proteins that are "downstream" to Msn2/4, such as the Mediator proteins Ssn3 and Gal11 and the NuA4 protein Eaf7. In addition, some deletions enable *HSP12-GFP*

induction via Msn2/4 alternative pathways that are repressed in the WT strain. Our analysis identifies the Msn2/4-dependent effect of each single knockout to evaluate the contribution of Msn2/4-regulating pathways on *HSP12* separately from alternative, parallel, or downstream factors.

### HOG pathway activity beyond osmotic stress conditions

The HOG pathway is primarily associated with response to high-osmolarity conditions. However, there are prior reports of possible HOG pathway involvement in other stress conditions (Winkler *et al*, 2002; Bilsland *et al*, 2004; Panadero *et al*, 2006). Our results show that the HOG pathway is modulating Hsp12-GFP in all the

conditions we tested. This might be evolutionary remains of HOG role as the key stress response pathway in the evolution of yeast (Gasch, 2007). There is a clear difference in HOG role in osmotic stress, where it has dramatic effect, and other conditions, where it is more of a fine-tuning modulator of the response. Some of the difference is due to the stress-specific role of Hot1 as an effector of the HOG pathway in osmotic stress (Fig 5D).

### Induction by the general stress response involves both transcriptional activation and de-repression

When discussing gene induction, such as *HSP12* stress-dependent induction, we need to consider a complementary repression

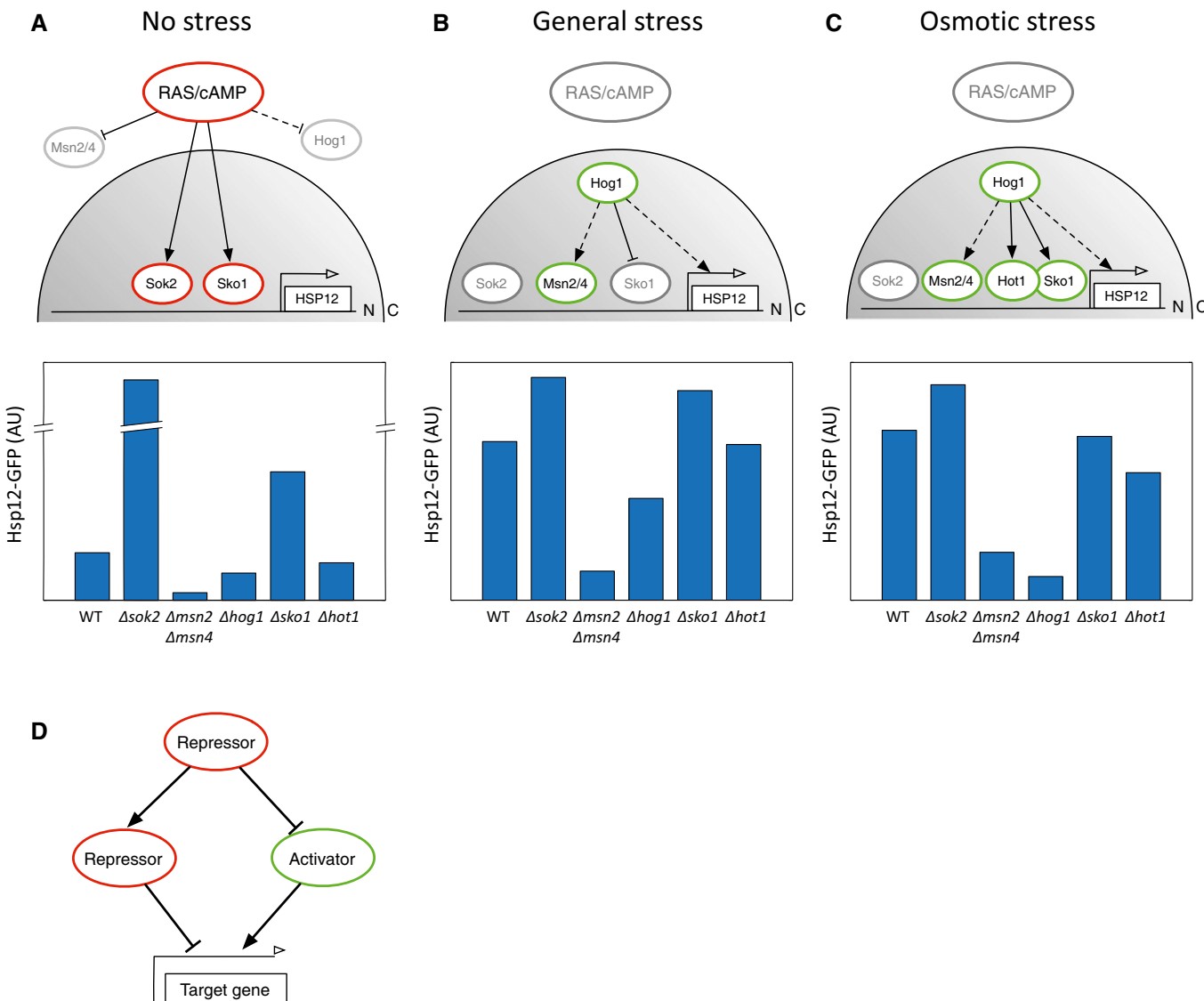

**Figure 6.  Induction of the general stress response by both activation and de-repression.**

A–C   Schematic illustration of *HSP12* regulation under different environmental conditions (top). Hsp12-GFP levels of the relevant knockout strains in log phase (A), redox stress (B), and osmotic stress (C) are shown (bottom).

D      Regulatory design pattern in the activation of the general stress response. The activation of a gene depends both on the de-repression of its direct activator and on the de-activation of its direct repressor. This network design contributes to the robustness in the induction and the termination of the response.

mechanism that ensures that the gene is not expressed in baseline states. We can distinguish between *passive* repression, which is insensitive to signals, and *regulated* repression, which is modulated by signals. An example of a passive repression mechanism might be generic mechanisms of nucleosome assembly that constantly attempt to assemble nucleosomes on accessible DNA regions. Such assembly can close open promoters, leading to subsequent repression. Passive repression serves to increase the activation barrier to suppress sporadic transcription. As such, induction of the gene requires mechanisms for bypassing the repression (e.g., nucleosome disassembly). In contrast, regulated repression is one that can be modulated by signals. Thus, it can strongly prohibit transcription in some conditions and allow transcription in others. One can envision the first as a barrier and the latter as a gate.

Most discussions of stress-dependent induction focus on the activating signal, implicitly assuming a passive repression model. In contrast, our results highlight the importance of regulated repression by Sok2 and Sko1. The stress response signal concurrently de-represses Sok2 and Sko1 and activates Msn2/4. Interestingly, de-repression mechanisms are tightly coordinated with activation: cAMP/PKA regulates both Sok2 and Msn2/4 localization, and the HOG pathway regulates the repression by Sko1 and activation by several transcription activators, including Msn2/4 (Fig 6).

This observation suggests a regulatory principle in which the signaling pathway at the same time de-represses and activates target genes to ensure a sharp transition from a tightly repressed state to a highly induced state (Fig 6D). Such a circuit has additional beneficial properties, including robustness—the same signal is responsible for both de-repression and activation, ensuring coordinated response—and reversibility—once the upstream signal is diminished, the active repression can rapidly turn off downstream effects. This design has interesting predictions on the dynamics of activation and repression when different components of the circuit are removed.

### Functional specialization of RAS paralogs and cross-pathway interactions

The budding yeast has two RAS paralogs, Ras1/2, both of which participate in regulating cAMP levels. The differences between them are unclear. It is known that their protein levels differ (Ghaemmaghami *et al*, 2003), suggesting temporal division of labor. In addition, it was suggested that there are differences in terms of interactions with other pathways (Hurwitz *et al*, 1995) and their role in stress response (Shama *et al*, 1998).

In our analysis, we see functional distinctions between the two. Deletion of *RAS2* had effects that are consistent with reduced cAMP production both in terms of the change in Msn2 activity and localization, and also in the interaction with other members of the pathway. In contrast, deletion of *RAS1* had different effects which seem only partially mediated through cAMP levels. This might be due to the redundancy with Ras2 that can compensate in controlling cAMP levels. The opposite is less likely as Ras1 protein levels are ~10-fold less than Ras2. We detect surprising interactions between *RAS1*, the HOG pathway, and the Mediator complex repressive functions, to further repress *HSP12* during mid-log growth. This type of crosstalk opens new avenues for understanding regulation of stress-responsive genes, and the switch between growth and stress. The high

conservation of these pathways between yeast and human suggests avenues for study in higher eukaryotes.

### Toward comprehensive reconstruction and quantitative analysis

In this project, we aimed to elucidate the signaling and transcriptional networks that regulate Msn2/4 activity. Our large-scale experimental strategy provided us with a broad view of the interplay between different components of this system and insights about specific parts of the system. However, it quickly became clear that we could not resolve the underlying pathways completely from our data alone. This is due to several reasons, some of which are more technical (indirect effects), and some are more fundamental (not all the double mutants were examined). We are still far from an automated process, mainly due to the multitude of biological phenomena exposed in different cellular and environmental conditions.

Despite these caveats, using our results we were able to reconstruct most of the known relationships investigated in decades of pinpointed research. Additionally, by layering stress response information, with nuclear localization and genetic interactions, we were able to identify novel and fundamental interactions in this highly studied system, demonstrating the power of our approach.

## Materials and Methods

### Strains and plasmids

Yeast strains used in this study are listed in Table EV2. The *HSP12-GFP* fusion strain was generated by genomic integration of PCR fragment amplified from *HSP12-GFP* strain from the yeast GFP collection (Huh *et al*, 2003) (Invitrogen) to the query strain YMS140α (strain #1), creating YMS140α *HSP12::HSP12-GFP-HIS3 MX6* (strain #2).

Single KOs query strains were created by replacing the complete *ORFs* with antibiotic resistance cassette (clonNAT resistance) amplified from pAG25 plasmid integrated by homologous recombination to #2 strain (strains 3–31). Strain #32 ($\Delta msn2\Delta msn4$) was generated by genomic integration of a PCR deletion fragment amplified from plasmid pAG32 (hygromycin resistance) into the *MSN4* loci in strain #3 ($\Delta msn2$). All genomic integrations were confirmed by PCR using appropriate oligonucleotides.

Double KO strains were generated using the synthetic genetic array method as previously described (Tong *et al*, 2001). Briefly, the query strains (2–31) were mated to the appropriate yeast deletion or DAmP Yeast Library (Breslow *et al*, 2008) (Open Biosystems) on yeast extract–peptone–dextrose (YPD) containing 10 g/l yeast extract, 20 g/l peptone, and 20 g/l dextrose, and diploids were selected on synthetic defined (SD) medium containing 6.7 g/l yeast nitrogen base, 20 g/l dextrose, 5.4 g/l $Na_2HPO_4$, 8.6 g/l $NaH_2PO_4*H_2O$, and dropout solution of amino acids −His +G418 +clonNAT. Diploids were then sporulated, and haploids were selected first on SD −His −Arg −Lys −Leu +canavanine +thialysine, and then a second round of haploid selection was performed on the same selection media. Haploids were further selected by two rounds of growth on SD −His −Arg −Lys −Leu +canavanine +thialysine +G418 +clonNAT media. For strains containing triple deletions, the same procedure was used, except that the query strain was #32,

diploid selection was done on YPED media +G418 +clonNAT, and final haploid selection was done on SD −His −Arg −Lys −Leu +canavanine +thialysine +G418 +clonNAT +hygromycin media. For strains containing single deletion, the same procedure was used, except that the query strain was #2, diploid selection was done on SD −His +G418 media, and final haploid selection was done on SD −His −Arg −Lys −Leu +canavanine +thialysine +G418 media.

Single KO strains used for Msn2 localization analysis were generated by co-transformation of the appropriate yeast deletion or DAmP yeast with the centromeric plasmid containing the *MSN2-GFP* fusion under the control of a constitutive ADH1 promoter and a *LEU2* marker (a gift from C. Schuller) (Görner *et al*, 1998) and centromeric TEF2p-mCherry-NLS cherry plasmid (URA3 marker) which was constructed by homologous recombination of PCR fragments containing TEF2p-NLS-mCherry fragment into PRS316 centromeric plasmid digested with BamHI and XhoI.

## Robotic growth protocol

In this project, we were dealing with a highly sensitive stress reporter to characterize the response of different mutant strains to environmental stress. To obtain reproducible baseline conditions for the experiment, we had a few key requirements from our growth protocol: long exponential growth prior to the flow cytometry experiment; no interventions during the final growth period; uniform treatment to all mutant strains, regardless of their growth rate; reaching a narrow OD range at the onset of the experiment; and high-throughput protocol that works with 96/384 microtiter well plates.

To meet those requirements, we developed a high-throughput robotic growth protocol. In the first step, yeast colonies grown on fresh agar plate were replicated to liquid (minimal growth medium (SD)) microtiter plate using a robotic colony copier (Singer Instruments RoToR). This replication was carried out through an intermediate liquid plate, using calibrated parameters, to minimize the initial OD values in the plate. The liquid plate was then loaded on a Tecan Freedom Evo 2000 liquid handling station. We input the necessary information regarding the specific plate, the timing of the different protocol phases, and the continuing flow cytometry experiment. From this point on, the protocol was fully automated.

The liquid plate was grown inside a robotic incubator at 30°C, for 16–24 h. Then, strains from the plate were inoculated to a new plate in a 1:50 ratio. After this step, we expected the OD values in the plate to be relatively homogeneous since the 16–24 h growth synchronized the OD values between the strains. The diluted plate was transferred to incubator for 5–7 h growth and the optical density of the wells was measured in a plate reader (Tecan Infinite F200). The plate was then split to four replicates with $X$:$2X$:$4X$:$8X$ dilution series, where $X$ is the chosen dilution constant returned by an optimization function that we designed. The function receives as an input the measured optical density of the data and the expected range of doubling times of the mutants in the plate (typically 1.5–3.5 h). The returned dilution factor is the one that will maximize the expected number of strains that will have at least one source in the desired optical density range at the beginning of the experiment.

The dilution plates were transferred into the incubator for a final growth period of 10 h. Then, the optical density of the four plates was measured, and for each well, we chose the source that best fits the desired optical density range. Wells for which no appropriate source was found remained empty. In practice, the percentage of success of the protocol was high, when on average 95% of the strains reached the flow cytometry experiment in the desired OD range. The strains that did not pass the protocol were usually extremely sick strains that had a very long doubling time or lag-phase. In some cases, strains failed to pass the protocol due to technical problems like bias in OD measurement and imprecise dilutions.

## Flow cytometry assaying of *HSP12-GFP* expression in different conditions

All samples were analyzed by high-throughput flow cytometry (BD FACSCalibur with CyTek upgrade) using the HyperCyt automated sampler (IntelliCyt). The HyperCyt sampler aspirates cell samples directly out of the 96-well microtiter plates and transfers them sequentially to the flow cytometer. Approximately 3 μl of sample was aspirated from each well, resulting in an average of ~5,000–10,000 cells per sample.

The different condition samples were prepared for measurement according to the following protocol. Strains that successfully passed the growth protocol were divided to four plates, which were exposed to different stress conditions and transferred to the flow cytometry station in a fully automated process. Logarithmic growth: fluorescence level was measured right after the division to four plates. Heat stress: the plate was inserted into an incubator (LiCONiC instruments) preheated to 37°C for 60 min prior to the measurement. Osmotic stress: the strain cultures were mixed with SD + KCl (1.6 M) to final concentration of 0.4 M KCl, 90 min before the measurement. Redox stress: the strain cultures were mixed with SD + diamide (10 mM) to a final concentration of 2.5 mM diamide, 90 min before the measurement.

The data of the post-diauxic shift condition were collected separately. The strains were replicated from agar plates to liquid SD media plate using a robotic colony copier (Singer Instruments RoToR). The replication was carried out through an intermediate liquid plate, to minimize the initial OD values in the final plate. The plates were transferred into the incubator (30°C) for 24 h and then transferred to the flow cytometry station.

## Flow cytometry data analysis

The data of each plate was partitioned into individual wells using a dynamic programming algorithm designed in the laboratory and implemented in MATLAB software (Mathworks, ver. 2012a). The data of each well were separately gated to remove dead cells, cell debris, and other non-typical events. The gating procedure that we developed filtered the cells by defining typical FSC and SSC limits for each population (Newman *et al*, 2006). Briefly, the scatter plot of FSC vs. SSC has a dense area that contains most of the cells and around it a sparse scattering of events that do not represent the population. Our procedure automatically identified the dense area and eliminated all the events outside of this area. The procedure received as an input the percent of cells that we would like to retain after the filtering (90% in this screen) and a resolution parameter, which is a trade-off between the accuracy of the procedure and its running time.

After gating, fluorescence intensity histograms of different strains were unimodal and resembled a log normal distribution (Fig EV1C). The median fluorescence of the cells was calculated and corrected for auto fluorescence by subtracting the median value of a strain without GFP tag. All experiments were done in 2–3 independent repeats, and the median fluorescence values showed high correlation between the repeats (Fig EV2A–C). The mean of the repeats was used for further analysis (Table EV3).

### Condition-specific effects

The magnitude of the response of a knockout strain to a specific condition was defined as the ratio between its median Hsp12-GFP levels to WT levels in the same condition. The knockout was defined to have an effect in a specific condition if the magnitude of its response was higher than 1.1 or lower than 0.8. Knockouts that had an effect in at least one of the stress conditions (heat, osmotic, or redox) were defined to have a general effect in stress. The magnitude of this effect was defined as the maximum between the magnitudes in the conditions the strains had an effect on. Knockouts that had an effect in only one of the compared conditions were defined to be specific to this condition. In case a knockout had an effect in more than one of the compared conditions, the magnitude of those effects was compared. If the difference between the magnitudes was larger than 0.3, the knockout was defined to be specific to the conditions in which the magnitude of the effect was higher than the average of the magnitudes (for knockouts that increase the response) or lower than the average (for knockouts that decrease the response).

### Msn2/4-dependent activity

We quantified the Msn2/4-dependent activity of knockout $X$ according to the following formula:

$$D_X = 1 - \left[ \frac{\text{GFP}_{\Delta msn2 \Delta msn4 \Delta X} - \text{GFP}_{\Delta msn2 \Delta msn4}}{\text{GFP}_{\Delta X}} \right]$$

where $\text{GFP}_{mut}$ are the Hsp12-GFP levels of strain mut. This value is 1 for knockouts whose effect on HSP12 is fully Msn2/4 dependent (in which case $\text{GFP}_{\Delta msn2 \Delta msn4 \Delta X} = \text{GFP}_{\Delta msn2 \Delta msn4}$). The value is smaller, $0 \leq D_X < 1$, for knockouts whose effect is partially Msn2/4 independent, specifically only $D_X * 100$ percent of this effect is Msn2/4 dependent. In contrast, $D_X > 1$ for knockouts of genes that work cooperatively with Msn2/4 in the activation of HSP12. Finally, $D_X < 0$ in case the deletion of Msn2/4 on the background of $\Delta X$ caused an increase in HSP12 activation (relative to $\Delta X$).

### Epistasis criteria

Generally, an epistasis interaction was identified in cases where the combination of two knockouts with different effects on Hsp12-GFP expression resembled the effect of one of the individual knockouts. This definition depends on thresholds that define how different/similar we expect the effects to be. Specifically, we only considered knockout pairs in which the difference between the effects of the knockouts was at least 20% of the maximum between the effects. We defined an epistasis in those pairs if one of the following criteria were met:

Pairs with opposite effects relative to WT:

$$\text{GFP}_{\Delta X} < 1.1 * \text{GFP}_{\text{WT}} \text{ and } \text{GFP}_{\Delta Y} > 1.1 * \text{GFP}_{\text{WT}}$$
$$\text{and } \text{GFP}_{\Delta X \Delta Y} > 0.8 * \text{GFP}_{\Delta Y} \tag{1}$$

$$\text{GFP}_{\Delta Y} < 0.8 * \text{GFP}_{\text{WT}} \text{ and } \text{GFP}_{\Delta X} > 0.8 * \text{GFP}_{\text{WT}}$$
$$\text{and } \text{GFP}_{\Delta X \Delta Y} < 1.1 * \text{GFP}_{\Delta Y} \tag{2}$$

Other:

$$|\text{GFP}_{\Delta X \Delta Y} - \text{GFP}_{\Delta Y}| < 0.2 * \max(\text{GFP}_{\Delta X}, \text{GFP}_{\Delta Y}) \text{ and}$$
$$|\text{GFP}_{\Delta X \Delta Y} - \text{GFP}_{\Delta X}| > 0.2 * \max(\text{GFP}_{\Delta X}, \text{GFP}_{\Delta Y}) \tag{3}$$

In each pair, both knockouts were considered once as $\Delta X$ and once as $\Delta Y$.

### Additive and multiplicative models

Following the additive definition, the effect of gene $X$ deletion was defined as: $F_{\Delta X} = \text{GFP}_{\Delta X} - \text{GFP}_{\text{WT}}$.

The expected value of $X, Y$ double deletion was calculated as: $\widehat{\text{GFP}}_{\Delta X \Delta Y} = F_{\Delta X} + F_{\Delta Y} + \text{GFP}_{\text{WT}}$.

According to the multiplicative definition, the effect of gene $X$ deletion was defined as: $F_{\Delta X} = \frac{\text{GFP}_{\Delta X}}{\text{GFP}_{\text{WT}}}$.

The expected value of $X, Y$ double deletion was calculated as: $\widehat{\text{GFP}}_{\Delta X \Delta Y} = F_{\Delta X} * F_{\Delta Y} * \text{GFP}_{\text{WT}}$.

The interaction in both cases was calculated as: $I_{\Delta X \Delta Y} = \widehat{\text{GFP}}_{\Delta X \Delta Y} - \text{GFP}_{\Delta X \Delta Y}$.

### Combined interaction model

In this interaction model, we combined the additive and the multiplicative definitions by assuming that each knockout has an additive ($F_{\Delta X}^+$) and a multiplicative ($F_{\Delta X}^*$) contribution to the phenotype.

The additive effect of gene $X$ deletion was defined as: $F_{\Delta X}^+ = \text{GFP}_{\Delta X}^+ - \text{GFP}_{\text{WT}}^+$.

The multiplicative effect of gene $X$ deletion was defined as: $F_{\Delta X}^* = \frac{\text{GFP}_{\Delta X}^*}{\text{GFP}_{\text{WT}}^*}$.

The expected value of the double KO was calculated as follows: $\widehat{\text{GFP}}_{\Delta X \Delta Y} = F_{\Delta X}^* * F_{\Delta Y}^* * \text{GFP}_{\text{WT}}^* + (F_{\Delta X}^+ + F_{\Delta Y}^+ + \text{GFP}_{\text{WT}}^+)$.

To decompose the measured effect of each knockout to its multiplicative and additive portions, we performed a constrained nonlinear optimization using the interior point algorithm (Waltz *et al*, 2005). The optimization was set to minimize the differences between the expected and the observed values of the double knockouts and to assure that the additive and multiplicative effects of each knockout will sum to its total measured effect (i.e., $\text{GFP}_{\Delta X} = \text{GFP}_{\Delta X}^+ + \text{GFP}_{\Delta X}^*$).

The interaction was calculated as: $I_{\Delta X \Delta Y} = \widehat{\text{GFP}}_{\Delta X \Delta Y} - \text{GFP}_{\Delta X \Delta Y}$.

### MSN2 localization analysis

Strains expressing Msn2-GFP and NLS-mCherry were grown overnight, diluted to OD $\sim 0.1$, and grown at OD < 0.6 for additional 7–8 h prior to stress application. The cells were then transferred to glass bottom plate (384 format, Matrical Biosciences) coated with concanavalin A. The cells were left to descend to the bottom of the plate for 25 min and then gently washed to remove cells not attached to the glass. Time-lapse microscopy of the cells in bright field, GFP, and RFP channels was taken with 7-min intervals for

~2 h following addition of stress using a scan-R high-content screening microscope (Olympus). All strains were tested under osmotic (0.4 M KCl) and redox stress (2.5 mM diamide) conditions.

Image analysis was done using in-lab developed MATLAB software. Briefly, cell borders were identified using the bright field images and nuclei were identified using the NLS-mCherry signal. The ratio between the nuclear and the cytoplasmic Msn2-GFP levels was calculated separately for each cell. We calculated the mean of this quantity over all cells in each image and plotted it over time (Fig 2E). Then, for each mutant, we calculated the area under the curve of this graph. We calculated this value for two independent biological repeats of each mutant (Fig EV4D). The final value that we use is the mean of the two repeats (Fig 2B, Table EV3).

Expanded View for this article is available online:
http://msb.embopress.org

## Acknowledgements
We thank A. Appleboim, D. Englberg, O.J. Rando, R. Sadeh, M. Schuldiner, and members of the Friedman laboratory for comments on this manuscript. We thank A. Brill-Klein and A. Weiner for helping establish the robotic experimental protocols. This work was supported in part by ERC grant 233169, ISF Center Grant 1796/12, and ISF I-CORE grant on "Chromatin and RNA in Gene Regulation".

## Author contributions
AS, JG, NF, AA and AR conceived and designed this study. JG and AS conducted the experiments with support from AR. JG and NF designed the computational methods. JG, AS and NF performed data analysis and wrote the manuscript with contributions from all authors.

## Conflict of interest
The authors declare that they have no conflict of interest.

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
