## [Review Process File · Molecular Systems Biology]

Condition-specific genetic interaction maps reveal crosstalk between the cAMP/PKA and the HOG MAPK pathways in the activation of the general stress response

Jenia Gutin, Amit Sadeh, Ayelet Rahat, Amir Aharoni and Nir Friedman

Corresponding author: Nir Friedman, Hebrew University

Review timeline:

Submission date:	19 July 2015
Editorial Decision:	21 August 2015
Revision received:	30 August 2015
Accepted:	11 September 2015

Editor: Maria Polychronidou

Transaction Report:

1st Editorial Decision

21 August 2015

Thank you again for submitting your work to Molecular Systems Biology. We have now heard back from two of the three referees who agreed to evaluate your manuscript. We are still expecting a report from reviewer #3, but since the recommendations of the other two referees are quite similar, I prefer to make a decision now rather than further delaying the process. If we receive comments from reviewer #3 we will forward them to you so that you can address any further issues raised. As you will see from the reports below, the referees think that the study is interesting and they acknowledge the quality and potential broader relevance of the presented data. However, they raise a series of -mostly minor- concerns, which should be carefully addressed in a revision of the manuscript. Since the referees' recommendations are rather clear, there is no need to repeat all the points listed below.

REFeree REPORTS

Reviewer #1:

The manuscript by Gutin et al reports results from high density genetic analysis coupled to high throughput data generation that address the pathways, components and interactions involved in stress signaling in yeast, particularly that mediated by the Msn2/Msn4 transcription factors. This is an appropriate focus of study since the yeast stress response serves as an excellent system for interrogating the interplay and integration of multiple signaling pathways in regulating a primary response of cells to their changing environment. The system is widely studied and results presented here will be of interest to a wide community, both in the signaling and the stress response fields.

The data generated by the investigators were both extensive and quite robust and the analysis of those data quite sophisticated and insightful. While the primary conclusion from this study is a confirmation of our preexisting understanding of the form and interplay of the pathways mediating the stress response, the investigators add a great deal of granularity to that network. For instance, the investigators identified components that affected transcriptional output independent of the nuclear entry of Msn2, the primary mode of regulating Msn2 activity. In addition, they documented crosstalk among the PKA pathway, the HOG pathway and mediator complex in the Msn2 stress response and pinpointed the loci for those cross connections. In sum, the investigators generated a trove of data on this complex pathway from which they extracted numerous novel insights and, moreover, these data will be of significant value to other investigators in the field and will likely prompt further studies.

On a side note, the investigators provide an intriguing approach to analyzing genetic interaction data. Comparing quantitative phenotypes from double mutants versus those of the individual single mutants can provide an indication of potential genetic interaction between the two genes, a key factor in developing network structures in a wide variety of situations. However, determining whether the double mutant phenotype is different than that expected from the phenotype of the individual mutants is dependent on how one determines what to expect in the case of no interaction. The consensus from lots of such experiments is that the expected phenotype of the double mutant in this null case is simply the product of the quantitative phenotypes of the single mutants. The investigators provided excellent data showing that this is true in some cases, but in other the expectation should be the additive values of the phenotypes and in others it should be both additive and multiplicative. This observation should resonate with investigators well beyond the stress response and signaling fields and might potentially prompt reevaluation of existing synthetic genetic interaction studies.

Minor comments:

Page 9, 1st full paragraph, 3rd line: "deletions that repress the cAMP pathway..." should be "deletions that activate ...".

Page 10 and Figures 4C and S5C: It's not clear whether the investigators mean to use R or R2 in the graphs and text.

Page 13, last paragraph: "Thus, suggesting ..." is a sentence fragment.

Figures 4 and 5: I didn't see where the authors defined "Row epistasis" and "Column epistasis." I assume that one means that the gene on the top row is epistatic to the gene in the right column and the other means the opposite, but I couldn't find that definition or the indication of which is which.

Reviewer #2:

In this manuscript, Gutin and colleagues describe a genetic strategy to dissect the regulation of a stress response. They have used of a fluorescent reporter system (Hsp12-GFP) as a proxy for the activation of the environmental stress response (ESR) and have automated the collection of measurements for this reporter for different genetic backgrounds in *S. cerevisiae*. Using this system they then measured ESR activation under 5 different conditions in a panel of 70 single and 1566 double knock-outs. For the 70 single knock-out strains they measured how much of the strain dependent changes were also depending on Msn2/4 (for the 5 conditions) and they also measured the nuclear translocation of Msn2 under 2 conditions. Using this information they could calculate measures of genetic-interactions for the double mutants. All together this data collection effort should provide with a very rich source of information on the regulatory interaction network that control the ESR under different growth conditions. The advantage of using a protein reporter system instead of general growth is that it allows for potentially higher accuracy in the measurements and also for an interpretation that is likely to be more directly connected to some of the genes under study. The authors then analysed their results to provide some further insight into the subfunctionalization of the *ras1/ras2* paralogs and the connections between the HOG and cAMP/PKA pathways and the Mediator complex. I find the work to be comprehensive and

interesting in many different aspects from the study of genetic interactions to the the specific interest in the stress response pathways of yeast. I have 2 very general comments/concerns and a few minor concerns. I think this work would be of broad interest.

Main comments/concerns

1 - The most interesting aspect of the this work is the attempt to dissect in an automated and large-scale way the regulatory network of a stress response. Cells have complex ways to integrate different environmental cues and the authors have managed here to collect in a very standardized way the contributions of single genes and combinations of pairs of genes to a very specific phenotype. Similar previous (conditional) genetic-interactions screens have been mostly done looking at growth phenotypes which are more indirect than the GFP readout picked here. However, it is somewhat disappointing that the interpretation of the results is done manually. As the authors admit towards the end of the discussion it was apparently not possible to put together these results algorithmically. I understand that this would not be easy to achieve but I think it would be interesting to know more about the limitations.

1.1 - Given the data, is it possible to define a set of rules that would convert the measurements into pathway models (e.g. a directed graph) ? If not, why do they fail ?

1.2 - How much would these models recover prior knowledge ?

1.3 - As a point for discussion, what do the authors think would be needed to be able to obtain a reconstruction of these regulatory networks ? Additional experimental data that would include conditional physical interactions, post-translational modification data ? Improvements in modelling methods ? Work in progress that the authors feel it should not be part of this manuscript ?

2 - Several cut-offs to determine conditional effects are not justified or benchmarked.

2.1 - The definition of a conditional effect of a KO is not justified or benchmarked. According to the methods section the authors compared the KO with the WT in a given condition and determined that a significant effect to be a GFP level that is higher than 1.1 or lower than 0.8 of the WT. One assumes that these cut-offs are related to a measured reproducibility of biological replicates of WT strains. The authors should show why these cut-offs were selected and if they are valid across all conditions. In particular it would be useful to know what is the variation of the measurement of replicates of WT in different conditions and what these cut-offs (below 0.8 and above 1.1) correspond to in terms of fraction of false-positive calls. In supplementary figure 1C the distributions of Hsp12 abundance for the WT (after KCL) appear to still overlap even with the *msn2/msn4* double KO. Would it not make more sense to use a cut-off that is independent of the WT distribution in a given condition such as a z-score or percentile ?

2.2 - Similar to the point above, in order to define condition specific effects of KOs the authors defined a cut-off of 0.3 (relative to the WT) as a significant difference between conditions. Again, this value is never justified. Is a 0.3 difference large relative to the sensitivity of the method and the biological variation in Hsp12 abundance in a given KO/condition pair ?

2.3 - As above, the criteria to define epistasis are also not justified.

Minor comments

- The log phase condition appears to be the most dissimilar condition in that a larger proportion of the single KOs tend to cause an increase in Hsp12 expression relative to the WT. This suggests that a larger number of genes work to inhibit Hsp12 expression in the log phase condition. Do the authors have a biological explanation for this ?

- There is a difference in the number of KOs shown in figure 1D (68 single KOs) and in figure 2 (60 single KOs). These are both different from the number listed in the text (70).

- The term epistasis is often used as an equivalent to genetic-interaction. In this manuscript the authors use the term to mean a specific genetic-interaction where the double mutant has the same phenotype as one of the single mutants. They use this to infer ordering of pathway events. Given that many readers might equate epistasis with the broader notion of genetic interaction it would be useful to make a clear definition at the start of the epistasis results section.

- The authors collected information for Hsp12 expression in double mutants in the 5 conditions but the similarity between the genetic interactions for these double mutants is not discussed at all in the manuscript. I noticed that the GFP values are broadly differently correlated across conditions but I did not have access to the genetic interaction scores. How similar are the genetic interaction scores

across conditions ? Is there a reason why the authors did not try to compute condition specific genetic interactions from these data ?

- There are several results that should be included in supplementary information. All quantitative values described in the text and figures should be available for future re-use. This includes: all of the single KO measurements (Hsp12-GFP levels, Msn2 AUCs, the dependence of Hsp12-GFP levels on Msn2/4); all of the computed values for the double KOs based on the Hsp12-GFP levels (e.g. epistasis scores, genetic interaction scores the additive/multiplicative factors inferred by genetic interaction models)

- I would suggest to put the explicit number of double knockouts in the abstract (1566 instead of "~1600")

1st Revision - authors' response

30 August 2015

Point-by-point response of reviewer comments

Reviewer #1:

The manuscript by Gutin et al reports results from high density genetic analysis coupled to high throughput data generation that address the pathways, components and interactions involved in stress signaling in yeast, particularly that mediated by the Msn2/Msn4 transcription factors. This is an appropriate focus of study since the yeast stress response serves as an excellent system for interrogating the interplay and integration of multiple signaling pathways in regulating a primary response of cells to their changing environment. The system is widely studied and results presented here will be of interest to a wide community, both in the signaling and the stress response fields.

The data generated by the investigators were both extensive and quite robust and the analysis of those data quite sophisticated and insightful. While the primary conclusion from this study is a confirmation of our preexisting understanding of the form and interplay of the pathways mediating the stress response, the investigators add a great deal of granularity to that network. For instance, the investigators identified components that affected transcriptional output independent of the nuclear entry of Msn2, the primary mode of regulating Msn2 activity. In addition, they documented crosstalk among the PKA pathway, the HOG pathway and mediator complex in the Msn2 stress response and pinpointed the loci for those cross connections. In sum, the investigators generated a trove of data on this complex pathway from which they extracted numerous novel insights and, moreover, these data will be of significant value to other investigators in the field and will likely prompt further studies.

We thank the reviewer for his/her support and kind evaluation of our manuscript.

On a side note, the investigators provide an intriguing approach to analyzing genetic interaction data. Comparing quantitative phenotypes from double mutants versus those of the individual single mutants can provide an indication of potential genetic interaction between the two genes, a key factor in developing network structures in a wide variety of situations. However, determining whether the double mutant phenotype is different than that expected from the phenotype of the individual mutants is dependent on how one determines what to expect in the case of no interaction. The consensus from lots of such experiments is that the expected phenotype of the double mutant in this null case is simply the product of the quantitative phenotypes of the single mutants. The investigators provided excellent data showing that this is true in some cases, but in other the expectation should be the additive values of the phenotypes and in others it should be both additive and multiplicative. This observation should resonate with investigators well beyond the stress response and signaling fields and might potentially prompt reevaluation of existing synthetic genetic interaction studies.

We are glad that we managed to emphasize this issue throughout the manuscript and thank the reviewer for the acknowledgment of its importance.

Minor comments:

Page 9, 1st full paragraph, 3rd line: "deletions that repress the cAMP pathway..." should be "deletions that activate ...".

We thank the reviewer for noticing this mistake. We revised the text accordingly.

Page 10 and Figures 4C and S5C: It's not clear whether the investigators mean to use R or R² in the graphs and text.

Our general guideline when working with those notations is using the R notation to quantify the strength of correlation between two independently measured data points (as in Figures S2 and 4). In Figure 5C we quantify the percent of variance explained ($=R^2$ by definition) by a model, thus we chose to use the R² notation in this case.

Page 13, last paragraph: "Thus, suggesting ..." is a sentence fragment.

We revised the text accordingly.

Figures 4 and 5: I didn't see where the authors defined "Row epistasis" and "Column epistasis." I assume that one means that the gene on the top row is epistatic to the gene in the right column and the other means the opposite, but I couldn't find that definition or the indication of which is which.

We thank the reviewer for pointing out this issue. We added an explicit definition of those indications to the figure caption.

Reviewer #2:

*In this manuscript, Gutin and colleagues describe a genetic strategy to dissect the regulation of a stress response. They have used of a fluorescent reporter system (Hsp12-GFP) as a proxy for the activation of the environmental stress response (ESR) and have automated the collection of measurements for this reporter for different genetic backgrounds in *S. cerevisiae*. Using this system they then measured ESR activation under 5 different conditions in a panel of 70 single and 1566 double knock-outs. For the 70 single knock-out strains they measured how much of the strain dependent changes were also depending on *Msn2/4* (for the 5 conditions) and they also measured the nuclear translocation of *Msn2* under 2 conditions. Using this information they could calculate measures of genetic-interactions for the double mutants. All together this data collection effort should provide with a very rich source of information on the regulatory interaction network that control the ESR under different growth conditions. The advantage of using a protein reporter system instead of general growth is that it allows for potentially higher accuracy in the measurements and also for an interpretation that is likely to be more directly connected to some of the genes under study. The authors then analysed their results to provide some further insight into the subfunctionalization of the *ras1/ras2* paralogs and the connections between the HOG and cAMP/PKA pathways and the Mediator complex. I find the work to be comprehensive and interesting in many different aspects from the study of genetic interactions to the the specific interest in the stress response pathways of yeast. I have 2 very general comments/concerns and a few minor concerns. I think this work would be of broad interest.*

We thank the reviewer for his/her support and kind evaluation of our manuscript.

Main comments/concerns

1 - The most interesting aspect of this work is the attempt to dissect in an automated and large-scale way the regulatory network of a stress response. Cells have complex ways to integrate different environmental cues and the authors have managed here to collect in a very standardized way the contributions of single genes and combinations of pairs of genes to a very specific phenotype. Similar previous (conditional) genetic-interactions screens have been mostly done looking at growth phenotypes which are more indirect than the GFP readout picked here. However, it is somewhat disappointing that the interpretation of the results is done manually. As the authors admit towards the end of the discussion it was apparently not possible to put together these results algorithmically. I understand that this would not be easy to achieve but I think it would be interesting to know more about the limitations.

1.1. - Given the data, is it possible to define a set of rules that would convert the measurements into pathway models (e.g. a directed graph) ? If not, why do they fail?

1.2 - How much would these models recover prior knowledge ?

1.3 - As a point for discussion, what do the authors think would be needed to be able to obtain a reconstruction of these regulatory networks ? Additional experimental data that would include conditional physical interactions, post-translational modification data ? Improvements in modelling methods ? Work in progress that the authors feel it should not be part of this manuscript ?

The reviewer touches on an important point that took much of our time during the analysis. The nature of publications is that we often tend not to discuss negative results and in this case one could always wonder whether the solution can come from a slightly different approach. At some point we decided we have to move forward with publication and left automatic construction on the sidelines. We agree with the reviewer that some insight as to what did not work would be important/interesting to the readers (especially ones with computational interests).

As we mention in the discussion we identify several aspects to the problem.

- The design of the experiment. Due to experimental constraints, our double expression matrix is not symmetric. Thus, there are relevant pairs of knockouts that do not appear in the matrix (e.g., we have A vs B and C vs B but not A vs C). Simple approach, such as drawing epistasis graph, suffers from omissions from unmeasured pairs. In retrospect there is a tradeoff between learning about additional knockouts (by making the matrix non-symmetric) vs completing all pairs. Many gene interaction studies made choices similar to ours in that many single KOs were crossed with a smaller number of “queries” to create a rectangular interaction matrix. This is good for some analyses but problematic for others.
- The nature of the system in question. The signaling pathways we study include many types of feedbacks at different levels. While we envision reconstruction using “A induce/repress B” type of arrows (such as the ones we show in our figure), manual intervention was needed to make an informed decision as to whether the effect is more likely due to secondary / feedback pathways rather than direct effect.
- This system is also different from others (e.g., chromatin biology) in that there are few complexes, and most proteins function alone. This means that uses of clustering to group statistically convincing reproduction of the same interactions fails.
- The type of relations we want to reconstruct. Finding linear cascades with alternative signs (e.g. A represses B, B repress C, ...) is easy with epistatic analysis. It becomes harder when we consider multiple inputs and output from a node, in which case simple epistasis is not useful.

We do not view the problem totally impossible, but we believe that it requires additional information (e.g., dynamics of the response, additional reporters, etc) to reliably reconstruct networks. We can of course add a network diagram (e.g., all epistasis edges) as a figure. But we view that as a partial reconstruction and not necessarily a trustworthy one. Since our focus is on understand the system (rather than methodology development) and given the large amount of findings we report, we decided to skip this.

We believe that making our data publically available would help further researchers who might have insights into the problem. We are now analyzing temporal dynamics of response in some of these mutants with the aim of distinguishing direct from indirect effects. Since these are in preliminary stage we do not mention them here.

2 - Several cut-offs to determine conditional effects are not justified or benchmarked.

2.1 - The definition of a conditional effect of a KO is not justified or benchmarked. According to the methods section the authors compared the KO with the WT in a given condition and determined that a significant effect to be a GFP level that is higher than 1.1 or lower than 0.8 of the WT. One assumes that these cut-offs are related to a measured reproducibility of biological replicates of WT strains. The authors should show why these cut-offs were selected and if they are valid across all conditions. In particular it would be useful to know what is the variation of the measurement of replicates of WT in different conditions and what these cut-offs (below 0.8 and above 1.1) correspond to in terms of fraction of false-positive calls. In supplementary figure 1C the distributions of Hsp12 abundance for the WT (after KCL) appear to still overlap even with the msn2/msn4 double KO. Would it not make more sense to use a cut-off that is independent of the WT distribution in a given condition such a z-score or percentile ?

2.2 - Similar to the point above, in order to define condition specific effects of KOs the authors defined a cut-off of 0.3 (relative to the WT) as a significant difference between conditions. Again, this value is never justified. Is a 0.3 difference large relative to the sensitivity of the method and the biological variation in Hsp12 abundance in a given KO/condition pair ?

2.3 - As above, the criteria to define epistasis are also not justified.

The reviewer raises an important point that shows up frequently during data analysis. On the one hand, using threshold makes it easier to report phenomena (numbers, venn diagrams, etc). On the other, any choice threshold is arbitrary to a certain extent (even if we use statistical theory to help us choose the 95% confidence interval, 95% is still an arbitrary choice). In such situation it is important to clarify two aspects. First, whether the results are robust to variation in the choice threshold, and second what degree of important one assigns to the discretization of the data.

We thus use threshold with some caution and treat threshold dependent results with a grain of salt. Specifically, changing the threshold for condition specific effect (points 2.1-2.2 above) would change to some extent the numbers on the Venn diagrams of Figure 1E, but would not change the trend (e.g., there are more Heat-specific effects than KCl- and Diamide- specific ones).

The choices of 0.8/1.1 represent our rough estimate of the variation of WT in our experiment. Thus, smaller changes might be due to technical variability. Again, most of our conclusions are robust to changing these specific choices.

Regarding the overlap the reviewer mentions, this is an overlap in the measurement of single cells within a population. Since these histograms are in log-scale, there is a substantial difference between the WT and msn2/4 KO.

Regarding the use of WT as a reference. This again was a point of much discussion among ourselves and with colleagues who commented on the paper. We believe that there is no absolute answer here, but find the comparison to WT as more appropriate. When we look for effects of a gene, we want to compare to some baseline, and the WT where all the components are present is a more natural one than some average of all strains we use in our experiment.

Minor comments

- The log phase condition appears to be the most dissimilar condition in that a larger

proportion of the single KOs tend to cause an increase in Hsp12 expression relative to the WT. This suggests that a larger number of genes work to inhibit Hsp12 expression in the log phase condition. Do the authors have a biological explanation for this ?

The reviewer raises an interesting point for which we had both biological and technical explanations. The technical explanation is that in optimal conditions the level of Hsp12 expression in WT cells is very low, thus some strains in which the expression level is slightly higher than WT in all conditions (by some additive constant) will be colored in a much brighter red in the log phase column (for example $\Delta cyr1$). The biological explanation is that some genes are not directly inhibiting the expression of *HSP12*, but their knockout induces non-optimal growth conditions, which are manifested by slower growth rate and activation of stress response. Possible examples for this phenomenon are the deletion of the gene encoding for a transcriptional activator of amino acid biosynthetic genes (*GCN4*), the deletions of proteasome related genes (*UMP1,UBP3,UBI4*) and the deletion of the gene encoding for superoxide dismutase (*SOD2*). We discuss this point in the results section in the revised manuscript.

- There is a difference in the number of KOs shown in figure 1D (68 single KOs) and in figure 2 (60 single KOs). These are both different from the number listed in the text (70).

We thank the reviewer for pointing out this issue, the correct number of single KOs is 68 and we revised the text accordingly. The number of records in Figure 2 represents the number of triple KO strains ($\Delta X \Delta msn2 \Delta msn4$) that we managed to construct (60 out of 65 possible).

- The term epistasis is often used as an equivalent to genetic-interaction. In this manuscript the authors use the term to mean a specific genetic-interaction where the double mutant has the same phenotype as one of the single mutants. They use this to infer ordering of pathway events. Given that many readers might equate epistasis with the broader notion of genetic interaction it would be useful to make a clear definition at the start of the epistasis results section.

We thank the reviewer for this comment. We use epistasis in the original and stricter meaning (the one going back to Bateson in the early 1900s). We agree that the distinction should be clarified. We revised the beginning of the relevant results section.

- The authors collected information for Hsp12 expression in double mutants in the 5 conditions but the similarity between the genetic interactions for these double mutants is not discussed at all in the manuscript. I noticed that the GFP values are broadly differently correlated across conditions but I did not have access to the genetic interaction scores. How similar are the genetic interaction scores across conditions? Is there a reason why the authors did not try to compute condition specific genetic interactions from these data?

We completely agree with the reviewer that the similarity of interactions between conditions is of interest. Throughout the analysis process we compared between interactions in different conditions, for example:

However many of the differential interactions that we observed were consistent with the conditional specific effect of the involved single KOs, and we did not see any global trends in the differences between the conditions. Clearly there are anecdotal examples that might be of great interest, however given the breadth of the manuscript, we decided not to elaborate on this aspect. We do examine a similar issue of the consistency of interactions between conditions in the epistasis interactions and we mentioned that most of those interactions are indeed consistent (top of Page 9).

- There are several results that should be included in supplementary information. All quantitative values described in the text and figures should be available for future re-use. This includes: all of the single KO measurements (Hsp12-GFP levels, Msn2 AUCs, the dependence of Hsp12-GFP levels on Msn2/4); all of the computed values for the double KOs based on the Hsp12-GFP levels (e.g. epistasis scores, genetic interaction scores the additive/multiplicative factors inferred by genetic interaction models)

We thank the reviewer for highlighting this issue. We added Extended View (supplementary) tables that include the single KO data (*HSP12-GFP* in five conditions, AUC in two conditions and the dependency on *Msn2/4* in five conditions). We expanded the double KO data table to contain the calculated epistasis and genetic interactions in all the measured conditions.

- I would suggest to put the explicit number of double knockouts in the abstract (1566 instead of "~1600")

We revised the text accordingly.